# Losing Dnmt3a dependent methylation in inhibitory neurons impairs neural function by a mechanism impacting Rett syndrome

Laura A Lavery[1,2], Kerstin Ure[1,2†], Ying-Wooi Wan[1,2], Chongyuan Luo[3,4], Alexander J Trostle[1,5], Wei Wang[1,2], Haijing Jin[6], Joanna Lopez[1,2], Jacinta Lucero[7], Mark A Durham[8,9], Rosa Castanon[3], Joseph R Nery[3], Zhandong Liu[1,2,6], Margaret Goodell[2,8,10,11,12], Joseph R Ecker[3,4], M Margarita Behrens[7,13], Huda Y Zoghbi[1,2,5,8,14,15]*

[1]Jan and Dan Duncan Neurological Research Institute at Texas Children's Hospital, Houston, United States; [2]Department of Molecular and Human Genetics, Baylor College of Medicine, Houston, United States; [3]Genomic Analysis Laboratory, The Salk Institute for Biological Studies, La Jolla, United States; [4]Howard Hughes Medical Institute, The Salk Institute for Biological Studies, La Jolla, United States; [5]Department of Pediatrics, Baylor College of Medicine, Houston, United States; [6]Graduate Program in Quantitative and Computational Biosciences, Baylor College of Medicine, Houston, United States; [7]Computational Neurobiology Laboratory, The Salk Institute for Biological Studies, La Jolla, United States; [8]Program in Developmental Biology, Baylor College of Medicine, Houston, United States; [9]Medical Scientist Training Program, Baylor College of Medicine, Houston, United States; [10]Center for Cell and Gene Therapy, Baylor College of Medicine, Houston, United States; [11]Stem Cells and Regenerative Medicine Center, Baylor College of Medicine, Houston, United States; [12]Department Molecular and Cellular Biology, Baylor College of Medicine, Houston, United States; [13]Department of Psychiatry, University of California San Diego, La Jolla, United States; [14]Department of Neuroscience, Baylor College of Medicine, Houston, United States; [15]Howard Hughes Medical Institute, Baylor College of Medicine, Houston, United States

*For correspondence:
hzoghbi@bcm.edu

Present address: †Animal Behavior Core, University of Ottawa, Ottawa, Canada

**Abstract** Methylated cytosine is an effector of epigenetic gene regulation. In the brain, Dnmt3a is the sole 'writer' of atypical non-CpG methylation (mCH), and MeCP2 is the only known 'reader' for mCH. We asked if MeCP2 is the sole reader for Dnmt3a dependent methylation by comparing mice lacking either protein in GABAergic inhibitory neurons. Loss of either protein causes overlapping and distinct features from the behavioral to molecular level. Loss of Dnmt3a causes global loss of mCH and a subset of mCG sites resulting in more widespread transcriptional alterations and severe neurological dysfunction than MeCP2 loss. These data suggest that MeCP2 is responsible for reading only part of the Dnmt3a dependent methylation in the brain. Importantly, the impact of MeCP2 on genes differentially expressed in both models shows a strong dependence on mCH, but not Dnmt3a dependent mCG, consistent with mCH playing a central role in the pathogenesis of Rett Syndrome.

## Introduction

DNA methylation is a covalent epigenetic mark deposited mainly on the C5-position of cytosine nucleotides in mammals. Although DNA methylation in mammals was described by Rollin Hotchkiss in 1948 (*Hotchkiss, 1948*), it took nearly thirty years to realize its function in the regulation of gene expression to shape fundamental biological processes including cell differentiation, genomic imprinting and genome stability (*Compere and Palmiter, 1981*; *Li and Zhang, 2014*).

Today many features of the classic model for gene regulation by DNA methylation have proven more complex. Rather than static and solely repressive to gene expression, DNA methylation is functionally dynamic (*Shen et al., 2014*), and in some cases has been shown to activate gene expression (*Aran et al., 2011*; *Ball et al., 2009*; *Hellman and Chess, 2007*; *Keown et al., 2017*) highlighting that context and protein factors that recognize methylated cytosine can alter the transcriptional outcome of this epigenetic mark.

It was also traditionally thought that DNA methylation in mammalian genomes could be found only at cytosines followed by guanine nucleotides (CpG methylation, or mCG), but it has become clear that cytosines in other contexts can also be methylated. This non-CpG methylation (also known as mCH, where H equals A, C or T) is not found in most mammalian cells but has been detected in pluripotent cells (*Lister et al., 2009*; *Ramsahoye et al., 2000*), some human tissues (*Schultz et al., 2015*), and is enriched in postnatal neurons (*He and Ecker, 2015*; *Lister et al., 2013*; *Price et al., 2019*). The mCH mark increases in neurons predominantly after birth and is concurrent with the period of peak synaptogenesis throughout the first 25 years of life in humans (or the first 4 weeks in mice). The resulting mCH pattern is largely conserved between mice and humans, where it makes up a high percentage of all DNA methylation in neurons (*He and Ecker, 2015*; *Lister et al., 2013*; *He et al., 2017*). Although the timing and specification of mCH suggest it serves a critical function in maturing neurons, the molecular mechanism and functional consequences for loss of the mCH mark remain understudied.

In mammals, DNA methylation is catalyzed by de novo DNA methyltransferases Dnmt3a and Dnmt3b, and maintained by Dnmt1 in the CpG context (*Jurkowska and Jeltsch, 2016*). Recent studies have shown that Dnmt3a is the sole 'writer' of mCH in postnatal neurons (*Lister et al., 2013*; *Gabel et al., 2015*; *Guo et al., 2014*; *Stroud et al., 2017*), where it is preferentially found in the CAC nucleotide context (*He and Ecker, 2015*). Mutations in *DNMT3A* have recently been associated with disorders of neurodevelopment (*Tatton-Brown et al., 2014*; *Sanders et al., 2015*; *Heyn et al., 2019*) suggesting that the mechanism by which Dnmt3a dependent methylation is written and read by other protein factors to regulate gene expression is critical for brain maturation. Loss of function of the only known 'reader' of mCH in the mammalian brain, Methyl-CpG-binding protein 2 (MeCP2) (*Gabel et al., 2015*; *Guo et al., 2014*; *Chen et al., 2015*), has long been associated with Rett syndrome (RTT) (*Amir et al., 1999*). RTT is an X-linked, postnatal neurological disorder; affected children are apparently healthy for the first 6–18 months of life, then lose their acquired milestones and develop a range of dysfunctions of the central and autonomic nervous systems (*Leonard et al., 2017*; *Zoghbi, 2016*). Mouse models of *MECP2* mutations in female mice faithfully recapitulate patient symptoms, the severity of which, in humans and mice, is determined by the specific mutation and the pattern of X-inactivation (*Leonard et al., 2017*; *Young and Zoghbi, 2004*; *Lombardi et al., 2015*). Numerous conditional knockout mice have revealed the importance of MeCP2 function in different cell populations in the brain as well as the etiology of RTT symptoms (*Chen et al., 2001*; *Gemelli et al., 2006*; *Fyffe et al., 2008*; *Samaco et al., 2009*; *Chao et al., 2010*; *Ito-Ishida et al., 2015*; *Meng et al., 2016*; *Zhang et al., 2014*; *Wang et al., 2014*; *Goffin et al., 2014*; *Ward et al., 2011*; *Adachi et al., 2009*; *Guy et al., 2001*; *Su et al., 2015*; *Johnson et al., 2017*). Given the apparent need of every neural cell type for MeCP2, it is not surprising that *MECP2* mutations have been linked to other neuropsychiatric conditions beyond RTT (*Zoghbi, 2016*; *Lombardi et al., 2015*). Yet re-expression of MeCP2 in the central nervous system (*Ross et al., 2016*) rescues even the severe neurological symptoms and the prematurely lethal phenotype of constitutive null male mice (*Chen et al., 2001*; *Guy et al., 2001*). RTT is thus driven by the loss of MeCP2 function in the nervous system, which continues to be essential into adulthood (*McGraw et al., 2011*; *Cheval et al., 2012*), but the underlying architecture is sufficiently intact to support functional rescue.

Despite decades of research, the precise mechanism by which MeCP2 drives RTT phenotypes remains enigmatic. The recent finding that MeCP2 binds to mCH at genes that are misregulated in

*Mecp2* knockout brains, and that mCH deposition occurs predominantly after birth in conjunction with neuronal maturation, has led to the hypothesis that failure of mutant MeCP2 to bind to mCH could account for the postnatal onset of RTT symptoms (*Chen et al., 2015*). Whether loss of MeCP2 binding to mCH is sufficient to cause RTT has not been explored.

Here we set out to test the hypothesis that MeCP2 is the only reader that interprets the unique methylation pattern set by Dnmt3a to direct gene expression in the maturing brain. If MeCP2 is the sole reader of these methylation marks, then loss of function of the writer should produce the same phenotype as loss of the reader at the behavioral and physiological level, as well as the same changes in gene expression at the molecular level. Such comparisons must be done in a cell-specific manner, since both mCH methylation (*Mo et al., 2015*) and the effects of MeCP2 loss are highly cell-specific (*Chen et al., 2001*; *Gemelli et al., 2006*; *Fyffe et al., 2008*; *Samaco et al., 2009*; *Chao et al., 2010*; *Ito-Ishida et al., 2015*; *Meng et al., 2016*; *Zhang et al., 2014*; *Wang et al., 2014*; *Goffin et al., 2014*; *Ward et al., 2011*; *Adachi et al., 2009*; *Guy et al., 2001*; *Su et al., 2015*; *Johnson et al., 2017*). No one has performed a cell-type specific head-to-head comparison of knockout models at the behavioral, physiological and molecular level, which is necessary to define the functional relationship between Dnmt3a and MeCP2 in the nervous system. In this study, therefore, we systematically compared the effects of cell-specific knockout for *Dnmt3a* and *Mecp2* in mice of the same genetic background at the behavioral, physiological, and molecular level. We chose GABAergic inhibitory neurons for this comparison because knockout of *Mecp2* in this cell type recapitulates most of the RTT-associated symptoms, including the delayed onset (*Chao et al., 2010*), and loss of Dnmt3a and the methylation pattern it sets in inhibitory neurons has not been previously studied. Intriguingly, our findings indicate that while mCH has significant contribution to RTT, MeCP2 is responsible for enacting only a subset of Dnmt3a dependent gene regulation. Together these data suggest there are other functional factors in this novel epigenetic pathway yet to be discovered that may impact neuropsychiatric phenotypes.

## Results

### Loss of Dnmt3a or MeCP2 in inhibitory neurons produces overlapping but not identical behavioral phenotypes

To compare the effects of loss of the mCH 'writer' and 'reader', we deleted *Dnmt3a* or *Mecp2* utilizing floxed alleles for each gene in combination with a mouse line that drives Cre expression in GABAergic inhibitory neurons (*Slc32a1-Cre, also known as Viaat-Cre*) (*Chen et al., 2001*; *Chao et al., 2010*; *Kaneda et al., 2004*). We chose the promoter for the *Slc32a1* gene to drive Cre expression because it has been shown to turn on in the central nervous system around embryonic day 10 (*Oh et al., 2005*), before the earliest mCH marks have been detected in the brain (~embryonic day 12.5 in the hindbrain) (*He et al., 2017*). We previously characterized mice with conditional knockout (cKO) of *Mecp2* in the same neurons on an F1 hybrid background (*Chao et al., 2010*), but to directly compare loss of the writer and reader we re-derived the *Mecp2* cKO on the C57BL/6J background. Conditional knockout mice showed loss of either Dnmt3a or MeCP2, respectively (*Figure 1—figure supplement 1*). We then tested both lines of mice for the same behavioral abnormalities previously reported (*Chao et al., 2010*).

Both cKO mice were born healthy but started to display symptoms around the time of weaning. *Table 1* summarizes the phenotypes, showing that there is considerable overlap in the phenotypes but also specific features that are present in only one of the models (see *Supplementary file 1* for statistics). While only males are classically tested for *Mecp2* knockout mice (*Mecp2* is on the X-chromosome), we tested both sexes for *Dnmt3a* cKO mice. We observed no significant difference between males and females in any tests and therefore combined data for all *Dnmt3a* cKO mice except for body weight because wild-type males are larger than females.

Both cKO mice first developed hind limb spasticity (*Figure 1A*), and by six weeks both *Dnmt3a* cKO and *Mecp2* cKO mice displayed obsessive grooming (*Figure 1B*), forepaw apraxia (*Figure 1C*), muscle weakness (*Figure 1D*), and motor deficits (*Figure 1E*, *Figure 1—figure supplement 2A*). We found hippocampal-dependent, but not amygdala-dependent, learning and memory deficits (*Figure 1F*, *Figure 1—figure supplement 2C*). Both mouse lines showed similar trends in anxiety-like behaviors (*Figure 1—figure supplement 2D–F*), with *Mecp2* cKO achieving significance in one

**Table 1.** Summary of behavioral and physiological test results in cKO models.

| Symptom Category | Test/observation | Result |
|---|---|---|
| General Health | Reduced body weight | *Dnmt3a* |
| | Prematurely moribund (humane end due to lesioning) | both |
| Spasticity | Hind limb clasping | both |
| Repetitive Behavior | Forepaw stereotypies | *Dnmt3a* |
| | Perseverative grooming and self-injury | both |
| Nociceptive pain | Delayed response to hot plate | both |
| | Delayed response to tail flick | *Dnmt3a* |
| Apraxia | Poor nest building | both |
| Muscle Strength | Decreased grip strength | both |
| Motor | Rotarod | none |
| | Parallel rod footslip- more footslips | *Mecp2* |
| | Open field – decreased distance traveled | both |
| | Open field – decreased vertical activity | both |
| Anxiety | Open field – time spent in center | none |
| | Elevated plus maze- increased time in open arms | *Mecp2* |
| | Light/dark box | none |
| Learning and Memory | Conditioned fear – decreased contextual fear response | both |
| | Conditioned fear – cued | none |
| | Conditioned fear – decreased response to tone #2 during fear learning (training) | *Dnmt3a* |
| Sensory processing | Increased acoustic startle response | *Dnmt3a* |
| | Paired pulse inhibition | none |
| Social Behavior | Partition | none |
| Inhibitory signaling | Altered miniature inhibitory postsynaptic currents | both |

test (*Figure 1—figure supplement 2F*). Starting at 7 to 8 weeks of age, the obsessive grooming led to skin lesions (*Figure 1G*); this self-injury was previously observed in *Mecp2* cKO mice on an F1 hybrid background (*Chao et al., 2010*) but was exacerbated on the C57BL/6J background, creating a need for humane euthanasia (see Materials and methods). This shortened the period we were able to perform behavioral testing on both cKO mouse models. Both cKO mouse models still responded to pain stimuli (*Figure 1—figure supplement 2G–H*), albeit to a lesser degree than controls. Rotarod and social interaction deficits that manifest in *Mecp2* cKO mice later in life on a F1 hybrid background did not develop within the foreshortened observation period (*Figure 1—figure supplement 2I–J*).

To compare cKO mouse models at the physiological level we measured miniature inhibitory postsynaptic currents (mIPSCs) in the dorsal striatum of 6 week old *Dnmt3a* cKO, *Mecp2* cKO, and control male mice. We observed decreased amplitude, as well as similar changes in frequency and charge (*Figure 1H–I*, *Figure 1—figure supplement 2K–L*). Differences were not apparent in measurements of rise or decay time (*Figure 1—figure supplement 2M–N*).

Despite these similarities, there were also clear differences between the cKO mice. *Dnmt3a* cKO mice were smaller than control mice starting at weaning age and throughout adulthood (*Figure 1J*). Forepaw stereotypies were evident in *Dnmt3a* cKO mice but not the *Mecp2* cKO mice on the C57BL/6J background (*Figure 1—video 1*). Loss of Dnmt3a in inhibitory neurons led to an increased acoustic startle response (*Figure 1K*), with loss of MeCP2 in the same cell type showing the opposite trend, similar to a previous MeCP2 study (*Chao et al., 2010*). Only the *Mecp2* cKO mice developed motor incoordination (*Figure 1L*) and showed a trend for increased sensorimotor gating (*Figure 1—figure supplement 2O*). *Dnmt3a* cKO mice also showed deficits in conditioned fear training (*Figure 1—figure supplement 2B*). Finally, the *Dnmt3a* cKO mice showed much more severe self-injury, requiring humane euthanasia 2.5 week earlier than *Mecp2* cKO mice (*Figure 1M*).

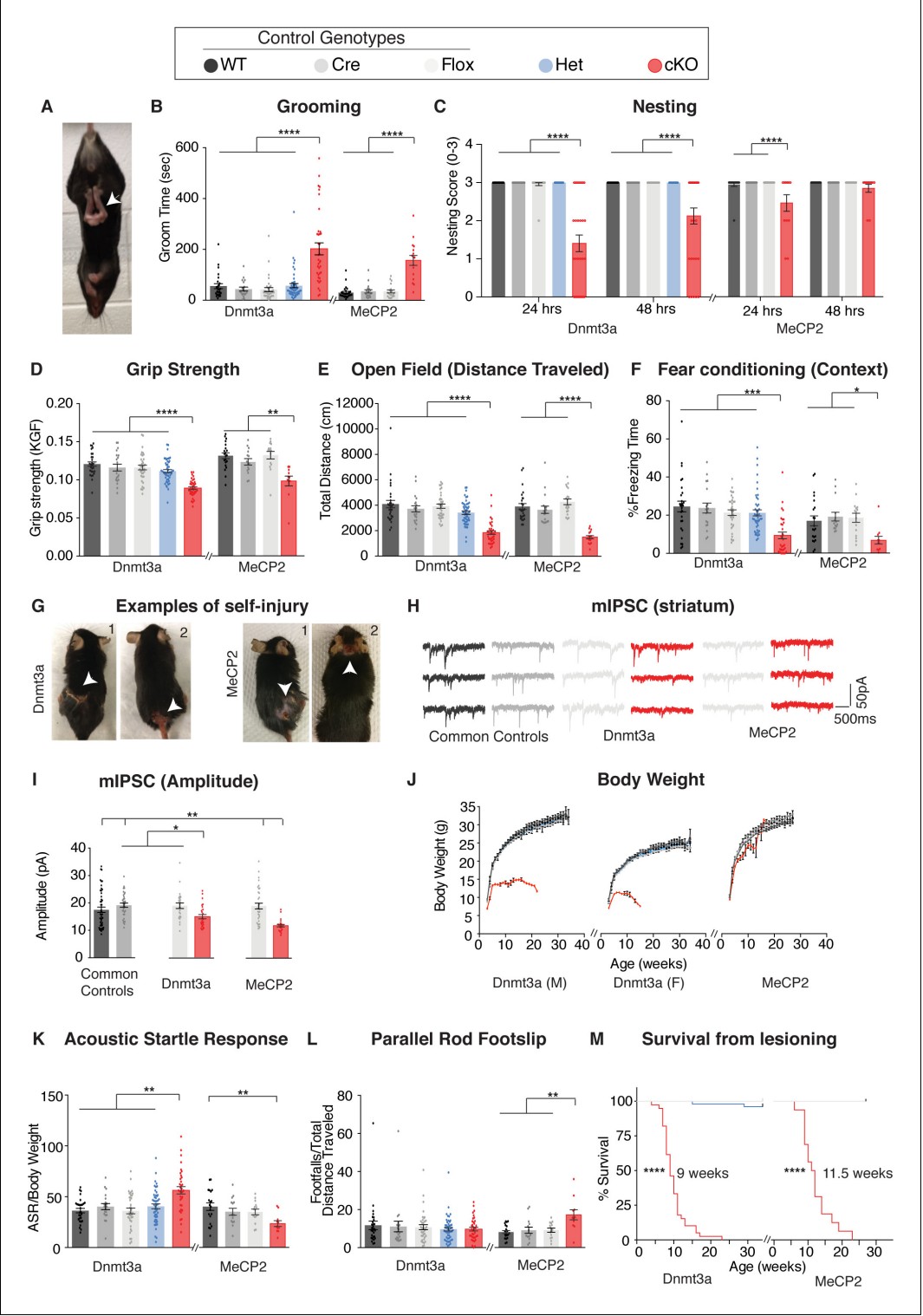

**Figure 1.** *Dnmt3a* cKO and *Mecp2* cKO mice show overlapping as well as distinct neurological deficits. (**A**) Mice that lack Dnmt3a or MeCP2 in inhibitory neurons present with hindlimb spasticity. (**B**) Obsessive grooming is increased in both cKO models. (**C**) Nest building, (**D**) grip strength, (**E**) open field, (**F**) fear conditioning tests revealed impairments in both cKO lines. (**G**) Self-injury in *Dnmt3a* cKO and *Mecp2* cKO mice necessitated humane euthanasia. (**H**) Example traces of miniature inhibitory postsynaptic currents (mIPSCs) recorded in the dorsal striatum. Both cKO models show similar alterations in (**I**) amplitude. (**J**) Weekly body weight records for *Dnmt3a* cKO and *Mecp2* cKO mice showed only *Dnmt3a* cKO mice (here separated by sex- see Materials and methods)
*Figure 1 continued on next page*

*Figure 1 continued*

were runted. (**K**) *Dnmt3a* cKO and *Mecp2* cKO mice showed opposite alterations in acoustic startle response. (**L**) Only *Mecp2* cKO mice displayed impairment on the parallel rod. **M**) *Dnmt3a* cKO mice had to undergo earlier euthanasia than *Mecp2* cKO mice due to the severity of their self-lesioning. n = 11–52 (behavior), n = 5–9 mice per genotype with 24–50 neurons total (electrophysiology). *p<0.05, **p<0.01, ***p<0.001, ****p<0.0001. See *Supplementary file 1* for full statistics.

The online version of this article includes the following video and figure supplement(s) for figure 1:

**Figure 1—figure supplement 1.** Protein levels of Dnmt3a and MeCP2 are reduced in inhibitory neurons of conditional knockout mice.

**Figure 1—figure supplement 2.** Supplemental behavioral and physiological data for *Dnmt3a* cKO and *Mecp2* cKO mice.

**Figure 1—figure supplement 3.** X-gal staining of *LacZ*$^{+/-}$;*Slc32a1-Cre*$^{+/-}$ and control E14.5 embryos shows widespread expression of Cre transgene in the nervous system and select expression in peripheral tissues.

**Figure 1—video 1.** Forepaw stereotypies are apparent in *Dnmt3a* cKO mice.

https://elifesciences.org/articles/52981#fig1video1

Of note we did observe sparse labeling of cells outside the nervous system using a *LacZ* reporter allele (*Soriano, 1999*) in combination with our *Slc32a1-Cre* driver (*Chao et al., 2010*; *Figure 1—figure supplement 3*). Given that MeCP2 function is largely restricted to the nervous system (*Chen et al., 2001*; *Guy et al., 2001*; *Ross et al., 2016*) and Dnmt3a has known peripheral functions (*Tatton-Brown et al., 2014*; *Heyn et al., 2019*; *Kaneda et al., 2004*; *Yang et al., 2015*; *Nishikawa et al., 2015*; *Okano et al., 1999*), we cannot rule out that non-neuronal symptoms (e.g. body weight) observed selectively in the *Dnmt3a* cKO mice may be in part due to loss of Dnmt3a in these select peripheral tissues.

## MeCP2 is a partial reader of Dnmt3a dependent methylation in striatal inhibitory neurons

Our behavioral data suggest that Dnmt3a and MeCP2 are functionally related, but it is still unclear what proportion of genes marked by Dnmt3a dependent methylation are regulated via MeCP2. To determine this, we tested our model that assumes MeCP2 is the only functional reader for these marks by sequencing the DNA methylome and transcriptome of sorted GABAergic inhibitory neuronal nuclei from the striatum of 6 week old *Dnmt3a* cKO, *Mecp2* cKO, and wild-type male mice via the INTACT method (*Mo et al., 2015*; *Figure 2A*). The striatum is ideal because mCH patterns are highly cell-specific and the vast majority of neurons (95%) in this region are inhibitory medium spiny neurons (*Dudman and Gerfen, 2015*; *Kemp and Powell, 1971*). We found that methylation was stable in the absence of MeCP2 (*Figure 2B*, *Figure 2—figure supplement 1*), but ~90% of mCH was lost in the absence of Dnmt3a (*Figure 2C*). We also noted a ~ 10% loss of mCG in *Dnmt3a* cKO neurons (*Figure 2D*) consistent with partial mCG demethylation in the prenatal period followed by re-methylation later in development (*He et al., 2017*; *Stroud et al., 2017*). Dnmt3a is thus required to re-methylate mCG sites in the postnatal period. To determine if mCH and mCG marks written by Dnmt3a are coincident at genomic loci or independent of one another, we plotted the percentage of mCH versus mCG per gene in wild-type mice, as well as restricting the analysis to Dnmt3a dependent methylation (the change in methylation observed in the *Dnmt3a* cKO is defined as 'Dnmt3a dependent'). While mCH and mCG in wild-type mice showed poor genome-wide correlation (*Figure 2E*), Dnmt3a dependent mCH and mCG showed significant correlation suggesting that the writing of mCH and mCG are coupled (*Figure 2F*).

Comparing gene expression by RNA-Seq for both cKO models, we found hundreds of differentially expressed genes (DEGs), either up- or down-regulated (*Figure 3A*, *Supplementary file 2*). Although a large portion of DEGs in the *Mecp2* cKO mice (~40%) were significantly altered in *Dnmt3a* cKO mice indicating a dependence on Dnmt3a and an important contribution to RTT, we were surprised to find that only a small percentage of DEGs in the *Dnmt3a* cKO mice (~12%) were significantly altered in *Mecp2* cKO mice (i.e. are 'MeCP2 dependent') (*Figure 3B*). The amount of overlap was consistent at different p-value thresholds, and whether we considered only the DEGs that were up-regulated, down-regulated, or of certain length (*Figure 3—figure supplement 1*, *Supplementary file 2*). Plotting the log2 fold-change in gene expression for each model against one

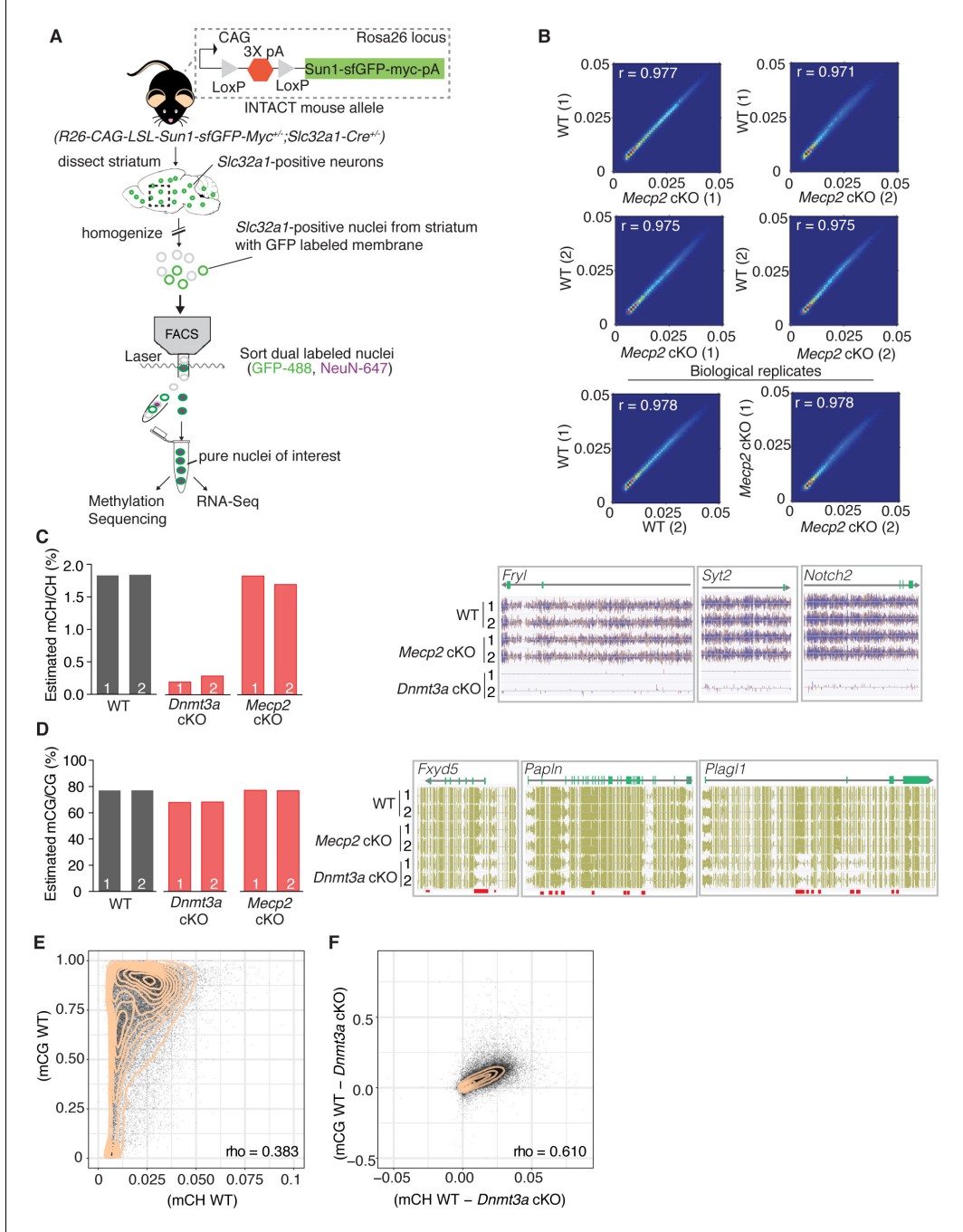

**Figure 2.** Methylation in striatal inhibitory neurons remains stable without MeCP2, but without Dnmt3a there is global loss of mCH and some loss of mCG. (**A**) Schematic of INTACT method used to isolate inhibitory neurons from the striata of WT, *Dnmt3a* cKO, or *Mecp2* cKO mice that also conditionally express the INTACT allele. (**B**) Spearman correlation of methylation profiles from inhibitory neurons sorted from WT vs. *Mecp2* cKO striatum, showing that methylation is stable in the absence of MeCP2. (**C**) Bar graph showing the global mCH level in each biological replicate (left). Example genes from DNA methylome sequencing tracks showing mCH signal in two biological replicates per genotype (right). (**D**) Bar graph showing the global mCG level in each biological replicate (left). Example genes from DNA methylome sequencing tracks showing mCG signal (right). Red bars indicate DMRs. (**E**) Genome wide correlation between mCH and mCG in wild-type mice showing poor correlation. (**F**) Genome wide correlation of Dnmt3a dependent mCG and mCH showing good correlation to indicate that mCH and mCG written by Dnmt3a are coupled. Correlation values for E and F are Pearson correlations designated as

*Figure 2 continued on next page*

*Figure 2 continued*

'rho'. n = 2 mice per genotype (see Materials and methods for specific genotype information). See **Supplementary file 1** for replicate statistics.

The online version of this article includes the following figure supplement(s) for figure 2:

**Figure 2—figure supplement 1.** Plot of the difference in genome wide mCH versus mCG methylation between the WT and *MeCP2* cKO mice.

---

another (*Figure 3C*), we found that DEGs common to both cKO models changed in the same direction and to a similar degree, consistent with an equal dependence on Dnmt3a and MeCP2. In all, our data show that MeCP2 is a restricted reader for Dnmt3a dependent methylation providing a platform for discovering novel functional factors in this pathway, as well as highlight a subset of genes regulated by MeCP2 that appear independent of Dnmt3a and the methylation it sets.

## Integrative gene expression and methylation analysis shows mCH and mCG loss contribute to *Dnmt3a* cKO DEGs, and reveals a strong mCH contribution to RTT

Plotting gene body mCH in wild-type mice for each category of differentially expressed genes, we find that, overall, DEGs have greater gene body mCH than genes that are unchanged, with up-regulated genes having higher mCH levels than down-regulated genes (*Figure 4A*, *Figure 4—figure supplement 1A*). The same analysis showed that mCG levels are also higher on DEGs than non-DEGs for the majority of DEGs categories (*Figure 4B*), consistant with the idea that total methylation may also influence the final transcriptional outcome (*Lagger et al., 2017*). Here we also see differences in mCG levels between up and down-regulated genes in each category, though the relationship between direction of gene expression change and higher mCG levels was not constant between DEG categories (*Figure 4B*, *Figure 4—figure supplement 1B*). *Figure 4—figure supplement 1C–F* shows gene body methylation levels for all genotypes and statistics for comparisons. Of note, some non-DEG genes show detectable changes in methylation (*Figure 4—figure supplement 1C–D*), however the magnitude of methylation loss is notably less than at DEG genes in either cKO model suggesting that the magnitude of methylation loss may drive the associated altered gene expression changes we observe.

To elucidate the relative contribution of mCH or mCG written by Dnmt3a to gene expression changes in our cKO models, we employed a similar method to previous publications (*Stroud et al., 2017*) and examined running average plots for the log2 fold-change in gene expression for each cKO model versus the percent change in Dnmt3a dependent methylation. These data were then fit with a univariate linear model to determine the percentage variance in log2 fold change explained by either mCH or mCG ($R^2$, see Materials and methods). For genes significantly misregulated only in the *Dnmt3a* cKO model we found a correlation with both Dnmt3a dependent mCH and mCG (*Figure 4C*), consistent with these marks being part of the same epigenetic program. These trends were statistically robust as when the same analysis was done 1000 times using random sets of non-DEG genes, the $R^2$ values for our selected set of DEGs were significantly higher (*Figure 4G*, see Materials and methods). The same trends hold when plotting the fold change in gene expression in the *Dnmt3a* cKO for genes that are commonly misregulated in both cKO models (*Figure 4D,H*). When we plot these commonly misregulated genes against the fold change in gene expression in the *MeCP2* cKO, we find a strong dependence on the change in gene expression to the change in mCH but not mCG (*Figure 4E,I*). As expected significant trends were not observed for Dnmt3a dependent methylation when plotting DEGs that were only significantly misregulated in the *MeCP2* cKO mice (*Figure 4F,J*).

As our RNA-seq data analysis could in theory detect DEGs that are specific to the minor portion of inhibitory interneurons that could co-purify with our sorted nuclei (<5% of neurons in the striatum) (*Dudman and Gerfen, 2015*; *Kemp and Powell, 1971*), we sought to identify DEGs that may be specific to interneurons of the mouse striatum to remove from our integrative analysis. We used Population-Specific Expression Analysis (PSEA) (*Kuhn et al., 2011*) to identify 38 candidate DEGs in our bulk RNA-seq dataset specific to interneurons (see Materials and methods). We excluded these genes from our integrative analysis and find that the trends and significance hold (*Figure 4—figure*

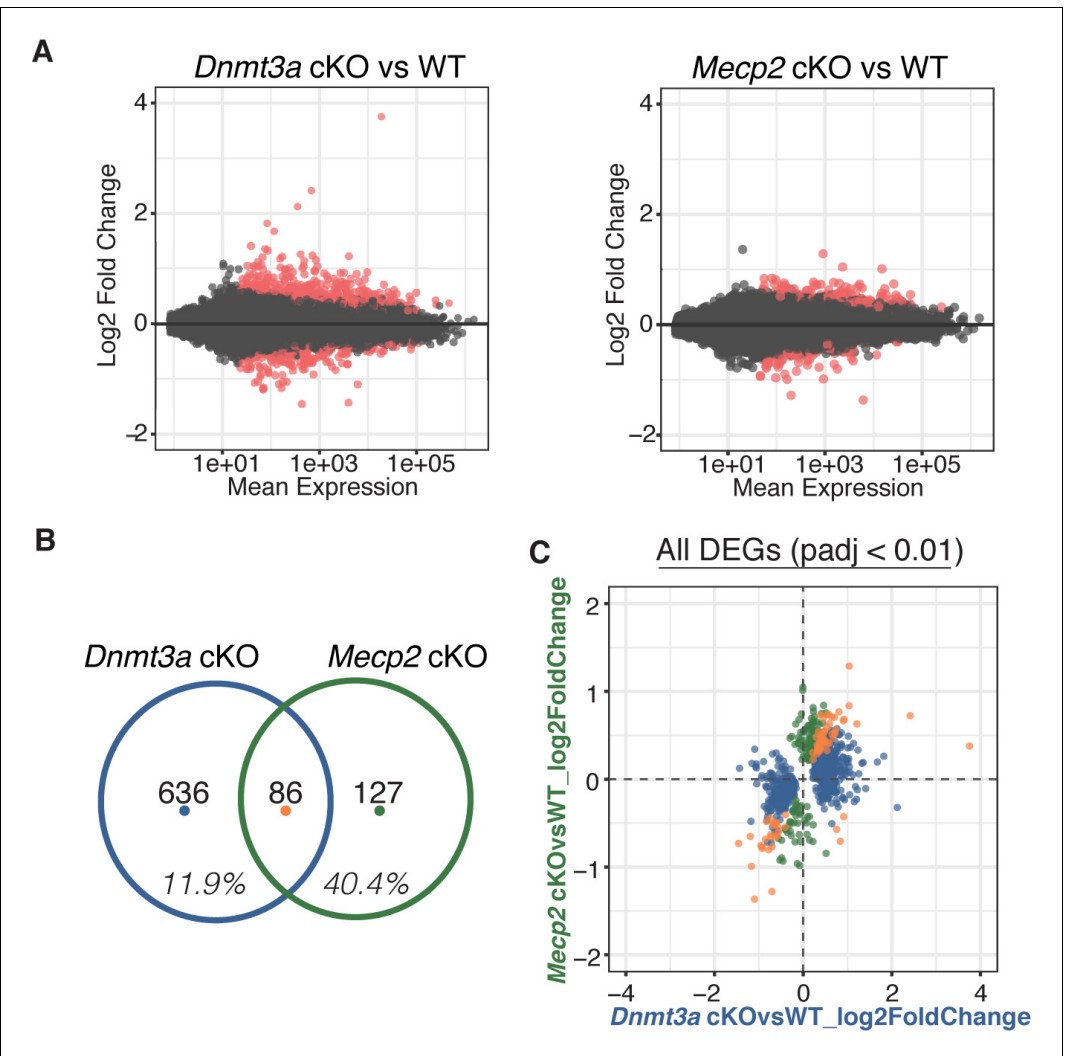

**Figure 3.** RNA-seq from sorted striatal inhibitory neurons of WT and cKO mice reveal MeCP2 is a restricted reader for Dnmt3a dependent methylation. (**A**) RNA-seq data of sorted WT vs *Dnmt3a* cKO or *Mecp2* cKO striatal inhibitory neurons that also express the INTACT allele. Red dots represent genes with altered expression in the knockout cells (padj <0.01). (**B**) Differentially expressed genes (DEGs) that overlap between knockout models. Inhibitory neurons that lack MeCP2 share ~40% of the same DEGs as the same neurons that lack Dnmt3a. Only ~12% of DEGs in inhibitory neurons that lack Dnmt3a are shared with the same neurons that lack MeCP2. (**C**) Plot of log2 fold-change for DEGs in *Dnmt3a* cKO and *Mecp2* cKO models. DEGs that are only significantly misregulated in the *Dnmt3a* cKO model, only significantly misregulated in the *Mecp2* cKO model, or common to both models are colored in blue, green, or orange, respectively. The plot shows that the DEGs common to both models have similar degree and direction of change.

The online version of this article includes the following figure supplement(s) for figure 3:

**Figure 3—figure supplement 1.** The percentage of DEGs that overlap between cKO mouse models broken down as a function of (**A**) p-value, (**B**) direction of change (down- or up-regulated), (**C**) gene length.

**Figure 3—figure supplement 2.** The percentage of DEGs that overlap between parvalbumin (PV) and vasoactive intestinal polypeptide (VIP) neurons in the mouse cortex from *Dnmt3a* cKO (Nestin-Cre) and *Mecp2* KO mouse models.

*supplement 2*). These analyses demonstrate that loss of Dnmt3a dependent mCH and mCG significantly contributes to gene expression changes seen in the *Dnmt3a* cKO. Importantly we find that mCH, but not Dnmt3a dependent mCG, does in fact play a central role in mediating RTT pathogenesis.

# Discussion

The discovery of mCH in postnatal neurons has shifted the view of how epigenetic gene regulation occurs in the brain. Beyond the obvious difference of unique context and cell-type restriction, the postnatal dynamics of this epigenetic program immediately suggest an important role for mCH in directing gene expression to ensure fully mature and functional neurons. Here we find that conditional knockout of the mCH writer, *Dnmt3a*, in GABAergic inhibitory neurons leads to genome-wide loss of mCH and a small subset of mCG sites, causing hundreds of gene expression changes to impair neurophysiology and behavior with some deficits mirrored in a matched deletion of *Mecp2*. We were surprised to find only a modest overlap of Dnmt3a/methylation dependent DEGs with MeCP2 dependent DEGs in our RNA-seq analysis. From this and our integrated gene expression and methylation analyses we distill two major conclusions. First, binding of MeCP2 to postnatal mCH contributes significantly to postnatal RTT symptoms, but there still appear to be other genomic targets recognized by MeCP2 or other functions whose loss contributes to the disease. Second, though MeCP2 'reads' a subset of Dnmt3a dependent methylation sites to significantly impact gene expression, there appears to be a much broader gene regulatory program set up by the placement of global mCH and some mCG via Dnmt3a. These findings highlight the need to compare gene function at multiple biological scales, the molecular level in particular, when drawing conclusions about the functional relationship between proteins.

Uncovering the relationship between Dnmt3a and MeCP2 in the nervous system is challenging for a several reasons. First, while the phenotypes observed in whole-body knockout of *Mecp2* are recapitulated in the brain-specific knockout (*Chen et al., 2001*; *Guy et al., 2001*), the phenotypes of whole-body (*Okano et al., 1999*) and brain-specific (*Nguyen et al., 2007*) knockout of *Dnmt3a* diverge, highlighting the essential role of Dnmt3a in peripheral tissues. In addition, while prior studies showed that early deletion of Dnmt3a in the brain drives neurological symptoms (*Nguyen et al., 2007*; *Kohno et al., 2014*), deletion of Dnmt3a using Cre-drivers that turn on postnatally have been demonstrated to produce minor (*Morris et al., 2014*) to no neurological symptoms (*Feng et al., 2010*). Therefore, it is likely that the underlying methylation pattern remains largely unaffected if Dnmt3a is deleted later in development, precluding the evolution of deficits. This is in contrast to MeCP2, where phenotypes are observed regardless of when the gene is deleted (*McGraw et al., 2011*; *Cheval et al., 2012*). Our study deliberately eliminated these variations and shows that Dnmt3a and MeCP2 have overlapping but distinct roles to produce functionally normal neurons and behaviors. These data highlight the importance of extensive direct comparison between knockout models in a selected cell type, on the same genetic background, and to consider the developmental contribution for each gene being compared.

While DNA methylation remains stable in the absence of MeCP2, we find that loss of Dnmt3a in inhibitory neurons before birth leads to global loss of mCH, with a minor drop in mCG (~10%), consistent with previous studies (*Lister et al., 2013*; *He et al., 2017*; *Stroud et al., 2017*). The loss of mCG is likely due to de-methylation in the embryonic brain (*Lister et al., 2013*; *He et al., 2017*) that cannot be re-methylated in the absence of Dnmt3a. Of note, while we interpret here the major driver of the phenotype observed in our *Dnmt3a* cKO mice is loss of the methylation Dnmt3a catalyzes during development, future experiments are necessary to determine if non-enzymatic functions of Dnmt3a may also contribute.

At the RNA level, we show that loss of either Dnmt3a or MeCP2 leads to misregulation of hundreds of genes that normally have higher methylation (mCH in particular) relative to genes that are unchanged. We find expression of these genes changes in both the up and down direction in a manner that is independent of gene length. Importantly, the vast majority of the genes misregulated in both cKO models change to a similar degree and in the same direction, consistent with equal dependence of these genes on Dnmt3a and MeCP2 for proper regulation. While our data cannot rule out the possibility that some of the DEGs we identify in our cKO mouse models are secondary, the percentage of overlapping DEGs between cKO models is robust to p-value cutoff, direction of change or gene length. Integrating our genomic datasets, we find a significant contribution of mCH and mCG to the fold change in gene expression in *Dnmt3a* cKO mice consistent with these marks functioning together to regulate gene expression during neuronal maturation. Intriguingly, we see a significant contribution of mCH, but not mCG to the misregulation of a subset of genes in the *MeCP2*

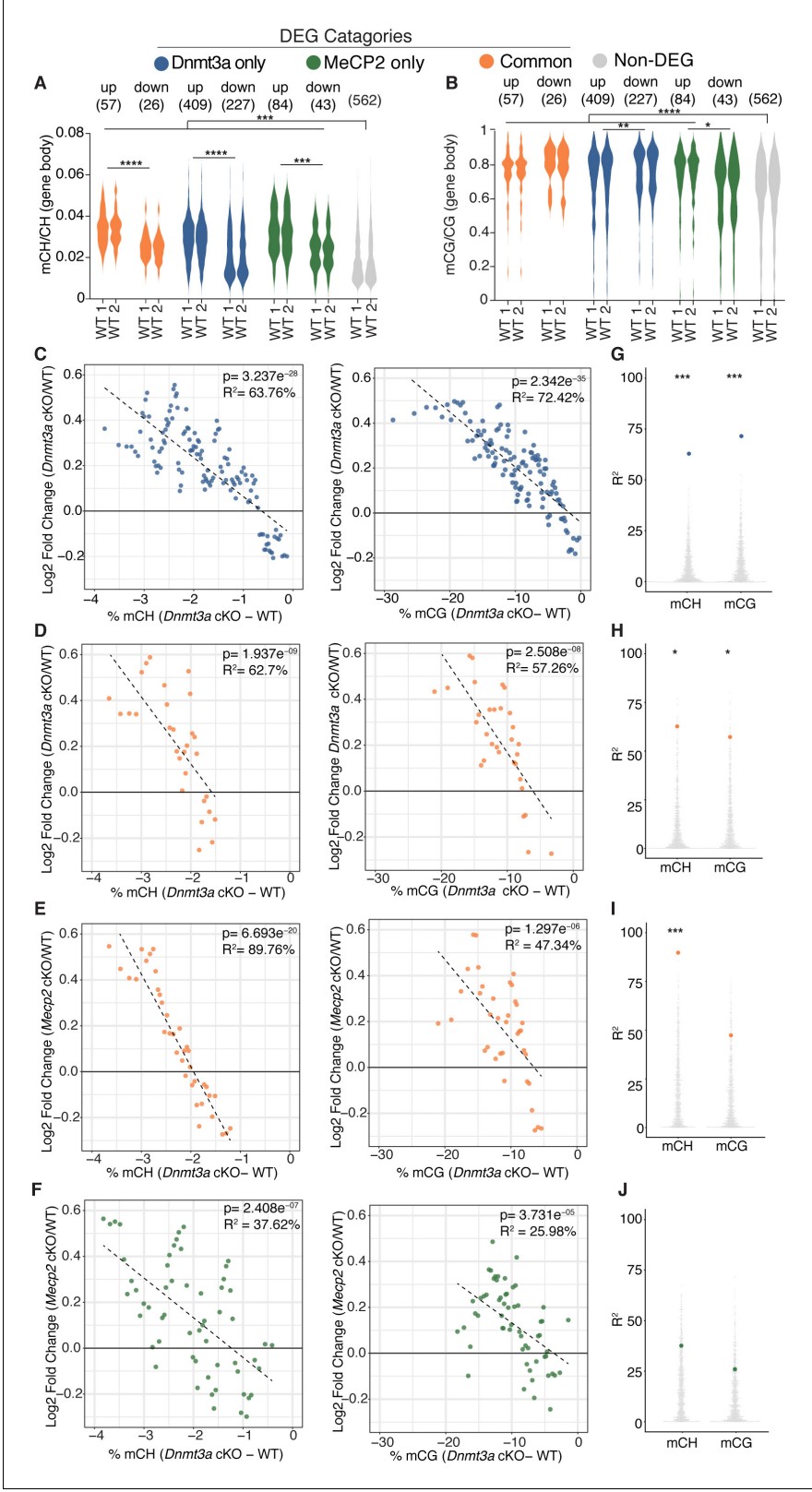

**Figure 4.** Integrative gene expression and methylation analysis shows mCH and mCG loss contribute to *Dnmt3a* cKO DEGs, and reveals a strong mCH contribution to RTT. (**A**) Gene-body mCH levels in different categories of DEGs in WT mice are plotted, highlighting that misregulated genes have higher mCH than genes that are unchanged and the significant differences between mCH levels on up- and down-regulated genes. (**B**) Gene-body

*Figure 4 continued on next page*

*Figure 4 continued*

mCG levels in different categories of DEGs in WT mice are plotted, demonstrating that misregulated genes (with the exception of MeCP2 down-regulated genes) have higher mCG than genes that are unchanged and the significant differences between mCG levels on up- and down-regulated genes. (C) Running average plot of log2fold change in gene expression for genes significantly misregulated only in the *Dnmt3a* cKO model versus the change in mCH methylation observed in the *Dnmt3a* cKO model ('Dnmt3a dependent mCH methylation') (left). Running average plot of log2fold change in gene expression for these same genes versus the change in mCG methylation observed in the Dnmt3a cKO ('Dnmt3a dependent mCG methylation') (right). (D) Running average plots of log2fold change in gene expression in the *Dnmt3a* cKO model for genes commonly misregulated in both cKO models versus Dnmt3a dependent mCH (right) and mCG (left). (E) Running average plots of log2fold change in gene expression in the *MeCP2* cKO for genes commonly misregulated in both cKO models versus Dnmt3a dependent mCH (right) and mCG (left). (F) Running average plots of log2fold change in gene expression for genes that are only significantly misregulated in the *MeCP2* cKO model. (G–J) $R^2$ values from analysis in panels C-F shown as blue, orange or green dots, respectively, plotted over 1000 random repetitions of the analysis with each repetition containing the same number of non-DEGs (padj >0.01). The results of random repetitions are shown as gray dots. All plots were made with DEGs padj <0.01. n = 2 mice per genotype for methylation data. n = 4 mice per genotype (RNA-seq). *p<0.05, **p<0.01, ***p<0.001, ****p<0.0001.

The online version of this article includes the following figure supplement(s) for figure 4:

**Figure 4—figure supplement 1.** Methylation levels on categories of DEG genes in WT, *Dnmt3a* cKO, and *Mecp2* cKO mice with statistical comparisons.

**Figure 4—figure supplement 2.** Integrative gene expression and methylation analysis excluding interneuron specific DEGs determined in PSEA analysis shows the same trends and statistical significance as in *Figure 4*.

---

cKO. This together with overlapping behavioral and physiological data suggests that binding of MeCP2 to mCH is critical for the postnatal evolution of RTT symptoms.

Our molecular data suggest that MeCP2 is likely one of a number of functional factors interpreting Dnmt3a dependent methylation. Though the overlap in gene expression changes between models has not been quantified until here, re-analysis of single-nuclear RNA-seq datasets from cortical inhibitory neurons (*Stroud et al., 2017*) are consistent with our conclusion (*Figure 3—figure supplement 2*). However, our data show a stronger dependence of *Mecp2* cKO DEGs on Dnmt3a function. With these data in hand we propose that the current model for Dnmt3a dependent gene regulation (*Stroud et al., 2017*) be updated to incorporate our insights that MeCP2 is a restricted reader for Dnmt3a dependent gene regulation, as well as that postnatal mCG plays a role in the epigenetic program driven by Dnmt3a during neuron maturation.

There is a defined set of proteins in the mammalian genome that possess a methyl binding domain (MBD) and have been shown to bind mCH in vitro (*Liu et al., 2018*). These data highlight these proteins as primary candidates, though further studies are necessary to determine whether the other MBD-containing proteins read mCH in the brain, in vivo. However, there are other proteins involved in transcriptional regulation that do not possess a classic MBD and are sensitive to the methylation status of DNA, with some validated in vivo (*Luo et al., 2018*; *Kribelbauer et al., 2017*; *Yin et al., 2017*; *Wu et al., 2010*; *Lomvardas and Maniatis, 2016*). For example, the DNA binding of many transcription factors is both positively and negatively affected by DNA methylation (*Kribelbauer et al., 2017*; *Yin et al., 2017*), a phenomena that is not exclusive to mammals (*O'Malley et al., 2016*). Further, the DNA methylation and histone modification machinery are intimately linked (*Lomvardas and Maniatis, 2016*). This link has been specifically shown in the differentiation of neural precursor cells, where methylation written by Dnmt3a antagonizes polycomb complexes to modulate gene expression (*Wu et al., 2010*). Given these findings, it is possible that similar mechanisms involving non-canonical methylation readers could be at play in Dnmt3a dependent gene regulation. Therefore, it will be important to undertake unbiased approaches to elucidate the full set of protein factors that drive gene regulation in this novel epigenetic pathway. All in all, our discovery that MeCP2 reads only some Dnmt3a dependent methylation marks lays the groundwork to uncover novel factors responsible for postnatal gene regulation that may be critical in setting up a mature, healthy mammalian brain.

# Materials and methods

**Key resources table**

| Reagent type (species) or resource | Designation | Source or reference | Identifiers | Additional Information |
|---|---|---|---|---|
| Antibody | Rabbit polyclonal anti-Dnmt3a | Santa Cruz Biotechnology | Cat. #: 20703; RRID:AB_2093990 | (1:1000) |
| Antibody | Rabbit polyclonal anti-MeCP2 | Huda Zoghbi lab (in house) | 0535 | (1:10,000) |
| Antibody | Rabbit polyclonal anti-histone H3 | Abcam | Cat. #: 1791; RRID:AB_302613 | (1:20,000) |
| Antibody | goat anti-rabbit-HRP | BioRad | Cat. #: 170–5046; RRID:AB_11125757 | (1:10,000) |
| Antibody | Mouse monoclonal anti-Dnmt3a | Novus | Cat. #: NB120-13888; RRID:AB_789607 | (1:250) |
| Antibody | Rabbit polyclonal anti-Myc | Sigma | Cat. #: C3956; RRID:AB_439680 | (1:200) |
| Antibody | Rabbit monoclonal anti-MeCP2 | Cell Signaling Technology | Cat. #: 3456; RRID:AB_2143849 | (1:500) |
| Antibody | Chicken polyclonal anti-GFP | Abcam | Cat. #: 13970; RRID:AB_300798 | (1:800) |
| Antibody | Goat anti-mouse Alexa Fluor 555 | Invitrogen | Cat. #: A21127; RRID:AB_141596 | (1:1000) |
| Antibody | Goat polyclonal anti-rabbit Dylight 488 | Bethyl Laboratories | Cat. #: A120-201D2; RRID:AB_10634085 | (1:750) |
| Antibody | Goat anti-chicken Alexa Fluor 488 | Invitrogen | Cat. #: A11039; RRID:AB_142924 | (1:750) |
| Antibody | Donkey anti-rabbit Alexa Fluor 568 | Thermo Fisher Scientific | Cat. #: A10042; RRID:AB_2534017 | (1:1000) |
| Antibody | Mouse monoclonal anti-NeuN | Millipore | Cat. #: MAB377; RRID:AB_2298772 | (1:300) (each batch needs to be empirically tested after labeling with Alexa Fluor 647) |
| Antibody | Rabbit polyclonal anti-GFP 488 | Thermo Fisher Scientific | Cat. #: A-21311; RRID:AB_221477 | (1:1000) |
| Other | cOmplete, EDTA-free Protease Inhibitor Cocktail | Sigma | Cat. #: 5056489001 | |
| Other | Pierce Universal Nuclease for Cell Lysis | Thermo Fisher Scientific | Cat. #: 88701 | |
| Other | Optimal Cutting Temperature medium | VWR | Cat. #: 25608–930 | |
| Other | VECTASHIELD HardSet Antifade Mounting Medium without DAPI | Vector Laboratories | Cat. #: H-1400 | |
| Other | Cryoseal XYL | Fisher Scientific | Cat. #: 22-050-262 | |
| Other | Aqueous Glutaraldehyde EM Grade, 10% 10 ML | Electron Microscopy Sciences | Cat. #: 16100 | Dilute fresh |
| Other | Paraformaldehyde | Sigma | Cat. #: P6148 | Make fresh |
| Other | X-Gal | Gold Biotechnologies | Cat. #: X4281C | |
| Other | RNasin Ribonuclease Inhibitors | Promega | Cat. #: N261A | |
| Other | UltraPure BSA (50 mg/mL) | Thermo Fisher Scientific | Cat. # AM2618 | |
| Other | DPBS, no calcium, no magnesium | Invitrogen | Cat. #: 14190144 | |

*Continued on next page*

*Continued*

| Reagent type (species) or resource | Designation | Source or reference | Identifiers | Additional Information |
|---|---|---|---|---|
| Other | UltraPure DEPC-Treated Water (1L) | Invitrogen | Cat. #: 750023 | |
| Commercial assay, kit | GE Healthcare Amersham ECL Prime Western Blotting Detection Reagent | Fisher Scientific | Cat. #: 45010090 | |
| Commercial assay, kit | APEX Alexa Fluor 647 Antibody Labeling Kit | Thermo Fisher Scientific | Cat. #: A10475 | |
| Commercial assay, kit | Single Cell RNA Purification Kit | Norgen Biotek | Cat. #: 51800 | |
| Commercial assay, kit | NuGEN Ovation RNA-Seq v2 | NuGen | protocol p/n 7102, kit p/n 7102–08 | |
| Commercial assay, kit | Rubicon ThruPlex DNA-Seq | Rubicon Genomics | protocol: QAM-108–002, kit p/n R400428 | |
| Commercial assay, kit | EZ DNA Methylation-Direct Kit | Zymo | Cat. #: D5021 | |
| Other | Unmethylated lambda DNA spike-in | Promega | Cat. #: D1521 | |
| Strain, Strain background (*Mus musculus*) | *Dnmt3a*$^{flox/flox}$ mice | Dr. Margaret Goodell, Baylor College of Medicine (Can be purchased from Riken BRC) | Cat. #: RBRC03731; RRID:IMSR_RBRC03731 | |
| Strain, Strain background (*Mus musculus*) | *Slc32a1-Cre*$^{+/+}$ mice | Jackson Laboratory | Cat. #: 017535; RRID:IMSR_JAX:017535 | backcrossed to C57Bl/6J |
| Strain, Strain background (*Mus musculus*) | *Mecp2*$^{flox/flox}$ and *Mecp2*$^{flox/y}$ mice | MMRRC | Cat. #: 011918-UCD; RRID:MMRRC_011918-UCD | backcrossed to C57Bl/6J |
| Strain, Strain background (*Mus musculus*) | R26-CAG-LSL-Sun1-sfGFP-Myc$^{+/+}$ mice | Dr. M. Margarita Behrens, The Salk Institute for Biological Studies (can be purchased from Jackson Laboratory) | Cat. #: 021039; RRID:IMSR_JAX:021039 | backcrossed to C57Bl/6J |
| Software, algorithm | GraphPad Prism version 6 | GraphPad Software | www.graphpad.com | |
| Other | pClamp10 | Molecular Devices | https://www.moleculardevices.com | |
| Software, algorithm | Minianalysis 6.0.3 | Synaptosoft Inc | http://www.synaptosoft.com/MiniAnalysis/ | |
| Software, algorithm | STAR aligner version 2.5.3a | GitHub | https://github.com/alexdobin/STAR | |
| Software, algorithm | FeatureCount v1.5.3 | Subread | http://subread.sourceforge.net | |
| Software, algorithm | DESeq2 v1.6.2 | Bioconductor | https://bioconductor.org/packages/release/bioc/html/DESeq2.html | |

## Animal husbandry and handling

Baylor College of Medicine Institutional Animal Care and Use Committee (IACUC, Protocol AN-1013) approved all mouse care and manipulation. Mice were housed in an AAALAS-certified Level three facility on a 14 hr light-10hr dark cycle. To obtain *Dnmt3a* cKO and control mice for behavior and electrophysiology, *Dnmt3a*$^{flox/flox}$ mice (**Kaneda et al., 2004**) (C57BL6 background) were bred to *Slc32a1-Cre*$^{+/+}$ mice (**Chao et al., 2010**) (C57BL/6J background) and *Dnmt3a*$^{flox/flox}$ mice were bred to wild-type mice (C57BL/6J background) to get F1 *Dnmt3a*$^{flox/+}$;*Slc32a1-Cre*$^{+/-}$ females and *Dnmt3a*$^{flox/+}$ males, respectively. F1 *Dnmt3a*$^{flox/+}$;*Slc32a1-*Cre$^{+/-}$ females were mated with *Dnmt3a*$^{flox/+}$ or *Dnmt3a*$^{flox/flox}$ males to obtain F2 mice (males and females) of the following genotypes for

behavior and electrophysiology: $Dnmt3a^{+/+}$ (WT), $Dnmt3a^{+/+};Slc32a1-Cre^{+/-}$ (Cre), $Dnmt3a^{flox/flox}$ (Flox), $Dnmt3a^{flox/+};Slc32a1-Cre^{+/-}$ (Het), $Dnmt3a^{flox/flox};Slc32a1-Cre^{+/-}$ (cKO). Importantly, female mice carrying the Cre allele at F1 were used in F2 mating as we discovered that males of the same genotype ($Dnmt3a^{flox/+};Slc32a1-Cre^{+/-}$) mated at F1 frequently transmit the Cre allele in the germline resulting in deletion of the $Dnmt3a$ allele in the whole body of F2 offspring. To obtain $Mecp2$ cKO and control mice for behavior and electrophysiology, $Mecp2^{flox/+}$ females (*Chen et al., 2001*) (C57BL/6J background) were mated to $Slc32a1-Cre^{+/-}$ males (C57BL/6J background) to get F1 male mice of the following genotypes: $Mecp2^{+/y}$ (WT), $Mecp2^{+/y};Slc32a1-Cre^{+/-}$ (Cre), $Mecp2^{flox/y}$ (Flox), $Mecp2^{flox/y};Slc32a1-Cre^{+/-}$ (cKO). Weekly weights and health scores were taken according to previously described disease score scale (*Guy et al., 2007*). For electrophysiological recordings control mice (WT and Cre) were shared for analysis and taken from both breeding schemes.

To obtain $Dnmt3a$ cKO, $Mecp2$ cKO and WT mice suitable for INTACT biochemistry experiments the breeding was as follows. $Dnmt3a^{flox/flox}$ mice were bred to $R26-CAG-LSL-Sun1-sfGFP-Myc^{+/+}$ mice (*Mo et al., 2015*) (C57BL/6J background) until both alleles were homozygosed to get $Dnmt3a^{flox/flox};R26-CAG-LSL-Sun1-sfGFP-Myc^{+/+}$ mice. $Dnmt3a^{flox/flox};R26-CAG-LSL-Sun1-sfGFP-Myc^{+/+}$ males were mated with $Dnmt3a^{flox/+};Slc32a1-Cre^{+/-}$ females to obtain $Dnmt3a^{flox/flox};R26-CAG-LSL-Sun1-sfGFP-Myc^{+/-};Slc32a1-Cre^{+/-}$ male mice used for biochemistry of striatal inhibitory neurons that lack Dnmt3a. $Mecp2^{flox/flox}$ females were mated to $R26-CAG-LSL-Sun1-sfGFP-Myc^{+/+}$ mice males until both alleles were homozygosed to obtain $Mecp2^{flox/flox};R26-CAG-LSL-Sun1-sfGFP-Myc^{+/+}$ female mice. $Mecp2^{flox/flox};R26-CAG-LSL-Sun1-sfGFP-Myc^{+/+}$ female mice were mated to $Slc32a1-Cre^{+/+}$ males to get $Mecp2^{flox/y};R26-CAG-LSL-Sun1-sfGFP-Myc^{+/-};Slc32a1-Cre^{+/-}$ male mice used for biochemistry of striatal inhibitory neurons that lack MeCP2. To obtain male mice used for biochemistry of wild-type striatal inhibitory neurons, $R26-CAG-LSL-Sun1-sfGFP-Myc^{+/+}$ mice were bred to $Slc32a1-Cre^{+/+}$ mice to get $R26-CAG-LSL-Sun1-sfGFP-Myc^{+/-};Slc32a1-Cre^{+/-}$ male mice. For all cohorts of mice, mixed genotypes were housed separated by sex up to six mice per cage until 6 weeks of age and reduced to no more than five per cage after 6 weeks.

Before experimental cohorts were subject to investigation we noted $Dnmt3a$ cKO and $Mecp2$ cKO mice had a self-injury phenotype leading to skin lesions. Therefore, the institutional veterinarian was consulted to define humane euthanasia endpoint criteria for further experiments. Mice with lesions could be treated with antibiotic cream, and mice were euthanized when lesions extended below the skin layer. Survival age was taken as the age in which a mouse died of natural causes or when a mouse was euthanized due to skin lesions according to the institutional veterinarian guidelines set beforehand. Investigators were blind to genotypes of experimental mice until behavioral tests were finalized.

## Mouse behavioral tests

$Dnmt3a$ cKO, $Mecp2$ cKO, and control mice were assigned individual ID numbers and the experimenters were blinded to genotype for the duration of behavioral testing. Experimental mice were divided into two cohorts for testing. Cohort one mice were subjected to the following tests, in order, at 6 weeks of age: open field, grooming, hotplate and tail flick, grip strength, parallel rod footslip, and prepulse inhibition/acoustic startle. The same mice were tested for conditioned fear at 8 weeks. Cohort two mice were subjected to the following tests, in order, at 6 weeks of age: elevated pus maze, light/dark box, rotarod, and partition/nesting. All behavioral assays were conducted during the light cycle, generally in the afternoon. All tests were conducted in light conditions.

All behavior data was analyzed using GraphPad Prism version 6 (GraphPad Software, La Jolla California USA, www.graphpad.com). Results were considered to be significant at $p < 0.05$ and statistical significance reported in figures using a star notation to represent the lowest significance reached in the comparison (*$p<0.05$, **$p<0.01$, ***$p<0.001$, ****$p<0.0001$). After behavioral analysis of the Dnmt3a cohort separated by both sex and genotype was completed, we determined that there was no statistical difference in phenotype between males and females. Therefore, data for male and female mice were merged in our final analysis with the exception of body weight measures, where wild-type males are larger than females. Statistical analysis for only males in the Dnmt3a cohort can also be found in *Supplementary file 1*.

## Grooming

Mice were habituated in the test room for 30 min. Each mouse was individually placed in a clean housing cage without food, food grate, or water for 10 min. They were then videotaped for an additional 10 min. After recording, the mice were returned to their home cage. Videos were scored for grooming time by an investigator blind to the genotype of the test mouse. Data is shown as mean ± standard error of mean and was analyzed by one-way ANOVA with Tukey's post hoc analysis.

## Grip strength

Mice were habituated in the test room for 30 min. Each mouse was allowed to grab the bar of a digital grip strength meter (Columbus Instruments, Columbus, OH) with both forepaws while being held by the tail and then pulled horizontally away from the meter with a constant slow force until the forepaws released. The grip (in kg of force) was recorded and the procedure repeated for a total of three pulls. Data shown is the average of the three pulls presented as mean ± standard error of mean. Grip strength was analyzed by one-way ANOVA with Tukey's post hoc analysis.

## Open field

Mice were habituated for 30 min in the test room lit at 200 lux with white noise playing at 60 dB. Each mouse was placed singly in the open field apparatus (OmniTech Electronics, Columbus, OH) and allowed to move freely for 30 min. Locomotion parameters and zones were recorded using Fusion activity monitoring software. Data is shown as mean ± standard error of mean and was analyzed by one-way ANOVA with Tukey's post hoc analysis.

## Fear conditioning

Mice were habituated for 30 min outside the test room. Mice were placed singly into the conditioned fear apparatus (Coulbourn Instruments, Holliston, MA) that consisted of a lighted box with a floor made of parallel metal bars. On the training day, mice were placed in the chamber and subjected to two rounds of training, each of which consisted of 180 s of silence followed by a 30 second-long 80–85 dB tone and 2 s of a 0.72 mA shock. 24 hr after training, the mice were returned to the box where they received the shock and freezing behavior recorded for six minutes (Context). One hour later, the grated floor of the test chamber was covered, the shape changed with plastic panels, and vanilla scent added to the chamber. Mice were returned to the apparatus and subjected to a cue test consisting of 180 s of silence followed by 180 s of the original 80–85 dB tone. Freezing behavior for all tests was scored using Freeze Frame three software (Actimetrics) with a threshold of 5.0. Data is shown as mean ± standard error of mean. Context tests were analyzed by one-way ANOVA with Tukey's post hoc analysis. Cue tests were analyzed by two-way ANOVA with Tukey's post hoc analysis.

## Hot plate and tail flick

Mice were habituated for 30 min in the test room prior to testing. Each mouse was placed individually on a 55℃ hot plate (Stoelting Co., Wood Dale, IL) and observed for jumping, vocalization, hind paw lifting, or licking of the hind paws. At the first incidence of any of these behaviors the mouse was removed from the hot plate and the elapsed time noted. 30 min after the hot plate test, the mouse was placed on the tail flick apparatus (Stoelting Co., Wood Dale, IL) and restrained with a paper towel laid over the mouse and held down gently by the experimenter's hand. The tail was laid in the groove above the lamp but was not restrained. The lamp was then turned on at 5-6eWatts; this also started the timer, which stopped automatically when the mouse moved its tail away from the lamp. The mouse was then returned to its home cage. Data was analyzed by one-way ANOVA with Tukey's post hoc analysis.

## Elevated plus maze

Mice were habituated for 30 min in the test room lit at 200 lux with white noise playing at 60 dB. The elevated plus maze is a plus sign-shaped maze with two opposite arms enclosed by walls and two opposite arms open without walls. The entire maze is elevated above the floor. Mice were placed singly at the intersection of the four arms and allowed to move freely for 10 min. Activity was

recorded by a suspended digital camera and recorded by the ANY-maze software (Stoelting Co., Wood Dale, IL). Data is shown as mean ± standard error of mean. Time and distance in the open arm were each analyzed by one-way ANOVA with Tukey's post hoc analysis.

### Light/Dark Box

Mice were habituated for 30 min in the test room lit at 200 lux with white noise playing at 60 dB. Mice were placed singly in the light side of the light dark apparatus (Omnitech Electronics, Columbus, OH) and allowed to move freely for 10 min. Locomotion parameters and zones were recorded using Fusion activity monitoring software. Data is shown as mean ± standard error of mean. Time in Light was analyzed by one-way ANOVA with Tukey's post hoc analysis.

### Parallel rod footslip

Mice were habituated in the test room for 30 min. Each mouse was placed in a footslip chamber consisting of a plexiglass box with a floor of parallel-positioned rods and allowed to move freely for 10 min. Movement was recorded by a suspended digital camera, while footslips were recorded using ANY-maze software (Stoelting Co.). At the completion of the test, mice were removed to their original home cage. Total footslips were normalized to the distance traveled for data analysis. Data is shown as mean ± standard error of mean and analyzed by one-way ANOVA with Tukey's post hoc analysis.

### Acoustic startle response and prepulse inhibition

Mice were habituated for 30 min outside the test room. Each mouse was placed singly in SR-LAB PPI apparatus (San Diego Instruments, San Diego, CA), which consisted of a Plexiglass cylindrical tube in a sound-insulated lighted box. Once restrained in the tube, the test mouse was allowed to habituate for 5 min with 70 dB white noise playing. The mouse was presented with eight types of stimulus, each presented six times in pseudo-random order with a 10–20 ms inter-trial period: no sound, a 40 ms 120 db startle burst, three 20 ms prepulse sounds of 74, 78, and 82 dB each presented alone, and a combination of each of the three prepulse intensities presented 100 ms before the 120 dB startle burst. After the test, mice were returned to their home cage. The acoustic startle response was recorded every 1 ms during the 65 ms period following the onset of the startle stimulus and was calculated as the average response to the 120 db startle burst normalized to body weight. Percent prepulse inhibition was calculated using the following formula: (1-(averaged startle response to prepulse before startle stimulus/averaged response to startle stimulus)) x 100. Data are shown as mean ± standard error of mean. Percent prepulse inhibition was analyzed by two-way ANOVA with Tukey's post hoc analysis, and acoustic startle response was analyzed by one-way ANOVA with Tukey's post hoc analysis.

### Rotarod

Mice were placed on the rotating cylinder of an accelerating rotarod apparatus (Ugo Basile, Varese, Italy) and allowed to move freely as the rotation increased from 5 rpm to 40 rpm over a five-minute period. Latency to fall was recorded when the mouse fell from the rod or when the mouse had ridden the rotating rod for two revolutions without regaining control. This procedure was repeated for a total of four trials for two days. Data is shown as mean ± standard error of mean. Latency to fall was analyzed by two-way ANOVA with Tukey's post hoc analysis.

### Partition test and nesting analysis

Mice were single-housed for 48 hr on one side of a standard housing cage. The cage was divided across its width by a divider with holes small enough to allow scent but no physical interaction. The test mouse was provided with a KimWipe folded in fourths as nesting material. At 24 hr and 48 hr of single-housing, the KimWipe was assessed for nesting score, as described previously (*Chao et al., 2010*). At least 16 hr before the partition test, a novel age- and gender-matched partner mouse of a different strain was placed on the opposite side of the partition. On the day of the test, the cage was placed on a well-lit flat surface. All nesting material, food pellets, and water bottles were removed from both sides of the cage, and the test mice were observed for 5 min while interaction time with the now-familiar partner mouse was recorded. Interactions involved the test mouse

smelling, chewing, or actively exploring the partition. At the end of the first test (Familiar 1), a novel mouse of the same age, gender, and strain replaced the familiar partner mouse, and test mouse interactions were recorded for five minutes (Novel). The novel mouse was then removed and the familiar partner mouse returned to the cage, followed by observation for another 5 min (Familiar 2). At the completion of the partition test, test mice were returned to their original home cage. Data is shown as mean ± standard error of mean. Interaction times were analyzed by two-way ANOVA with Tukey's post hoc analysis, and nesting scores were analyzed by one-way ANOVA with Tukey's post hoc analysis.

## X-gal staining of whole E14.5 embryos

X-gal staining was done as in *Hurd et al. (2007)*. Specifically, timed matings between *Slc32a1-Cre*$^{+/-}$ (JAX#017535) (*Chao et al., 2010*) and *LacZ*$^{+/+}$ mice (JAX#003474) (*Soriano, 1999*) were set, E14.5 embryos were dissected and fixed in a 0.5% formalin solution (0.5% formaldehyde, 2 mM MgCl$_2$, 5 mM EGTA, in 1X PBS) overnight at 4°C. Embryos were transferred to a 30% sucrose solution containing 2 mM MgCl$_2$ and incubated at 4°C overnight. Fixed, cryoprotected embryos were embedded in Optimal Cutting Temperature medium (VWR 25608–930) and stored at 80°C. Sagittal sections of *LacZ*$^{+/-}$;*Slc32a1-Cre*$^{+/-}$ or *LacZ+/-* control mice were processed at 25 m on a Leica CM3050S cryostat, dried on slides and stored at 20°C prior to staining. For X-gal staining, slides containing sections of *LacZ*$^{+/-}$;*Slc32a1-Cre*$^{+/-}$ and control mice were equilibrated at room temperature for 20 min. Slides were incubated for 5 min at room temperature in a solution containing 0.5% glutaraldehyde, 1.25 mM EGTA, 2 mM MgCl$_2$ in 1X PBS. Slides were then washed 3 times for 5 min each in wash buffer (2 mM MgCl$_2$, 0.02% IGEPAL-630, 0.1M Sodium Phosphate buffer pH 7.4) and stained at 37°C in X-gal staining buffer (5 mM $_{K4}$(Fe(CN)$_6$), 5 mM $_{K3}$(Fe(CN)$_6$), 1 mg/mL X-gal, 2 mM MgCl$_2$, 0.02% IGE-PAL-630, 0.1M Sodium Phosphate buffer pH 7.4) until blue pigment was well detected on slides with sections from *LacZ*$^{+/-}$;*Slc32a1-Cre*$^{+/-}$ (less than 2 hr). Slides were washed again in wash buffer 3 times for 5 min, 1X PBS for 5 min, and fixed with 4% Paraformaldehyde for 5 min as room temperature. After fixation slides were washed in 1X PBS 2 times for 5 min each and rinsed in milliQ water for 5 min. Slides were counterstained in eosin for 2–3 sec, washed with 95% ethanol 2 times 30 sec each, 100% ethanol 2 times 30 sec each, xylene 2 times for 30 sec each and mounted in Cryoseal XYL (Fisher Scientific Cat# 22-050-262). Final slides were imaged at 20X on a Axio Scan.Z1. A pathology report for unbiased identification of specific regions of X-gal staining was obtained from the Research Services Laboratory at the Center for Comparative Medicine at Baylor College of Medicine.

## Western blots

*Dnmt3a*$^{flox/flox}$;*R26-CAG-LSL-Sun1-sfGFP-Myc*$^{+/-}$;*Slc32a1-Cre*$^{+/-}$, *R26-CAG-LSL-Sun1-sfGFP-Myc*$^{+/-}$; *Slc32a1-Cre*$^{+/-}$ and *Mecp2*$^{flox/Y}$;*R26-CAG-LSL-Sun1-sfGFP-Myc*$^{+/-}$;*Slc32a1-Cre*$^{+/-}$ male mice, 6–16 weeks old were anaesthetize with isoflurane, whole brains dissected, split in half, frozen in liquid nitrogen and stored at 80°C till processing. Half-brains were thawed in 2 mL lysis buffer (20 mM HEPES pH 7.4, 200 mM NaCl, 100 mM Na$_3$PO$_4$, 1% Triton X-100, 1X protease inhibitor (Sigma Cat#5056489001), 1:2000 Universal Nuclease (Sigma Cat# 88701), dounced 50X with tight pestle and incubated on ice for 20 min. Lysates were spun down 2  20 min at 13,200 rpm at 4°C saving the supernatant each time. 20 g of protein was loaded onto a NuPAGE 4–12% Bis-Tris gradient gel and run in MES Running Buffer (Thermo Fisher Cat#NP0321BOX and NP000202, respectively). Protein bands were transferred to a nitrocellulose membrane in Tris-Glycine buffer (25mM Tris, 192mM glycine) plus 10% methanol at 400mA for 1 hr at 4°C. Membrane was blocked with 5% milk in tris buffered saline (5mM Tris pH 7.5, 120mM NaCl) with 0.1% Tween-20 (TBST) for 1 hr at room temperature (RT), cut and appropriate segments stained with primary antibodies overnight at 4°C. Membranes were washed in 1X TBST 3x for 5 min at RT and stained with secondary antibodies at 4°C for 1 hr, followed by repeated washes in 1X TBST. ECL detection kit (Fisher 45010090) was used to detect HRP. Antibodies used were: 1:1000 anti-Dnmt3a (Santa Cruz 20703), 1:10,000 anti-MeCP2 (in house N-terminal antibody), 1:20,000 anti-histone H3 (Abcam 1791) for primary antibodies, and 1:10,000 goat anti-rabbit-HRP (BioRad 170–5046) for the secondary antibody. Data is shown as mean ± standard error of mean and subject to unpaired t test with Welch's correction for statistical analysis.

## Immunofluorescence

2 week old *Dnmt3a*[flox/flox];*R26-CAG-LSL-Sun1-sfGFP-Myc*[+/-];*Slc32a1-Cre*[+/-] and *R26-CAG-LSL-Sun1-sfGFP-Myc*[+/-];*Slc32a1-Cre*[+/-] brains were dissected and fixed in PBS-buffered 4% paraformaldehyde overnight at 4°C, equilibrated in 30% sucrose for 2 days and then frozen in Optimal Cutting Temperature medium (VWR 25608–930). 6 week old *Mecp2*[flox/Y];*R26-CAG-LSL-Sun1-sfGFP-Myc*[+/-];*Slc32a1-Cre*[+/-] and *R26-CAG-LSL-Sun1-sfGFP-Myc*[+/-];*Slc32a1-Cre*[+/-] male mice were give Buprenorphine 30 min before being anaesthetize with Rodent Combo III and subject to transcardial perfusion using ice cold 1XPBS followed by PBS-buffered 4% paraformaldehyde, 10 mL each. Brains were post-fixed in PBS-buffered 4% paraformaldehyde overnight at 4°C, equilibrated in 30% sucrose for 2 days and then frozen in Optimal Cutting Temperature medium (VWR 25608–930). For 2 week old brains, 25 m sagittal sections were taken on a Leica CM3050S cryostat, dried on slides and washed briefly in 1XPBS. Slides were transferred to antigen retrieval buffer (10 mM Citrate pH6, 0.05% Tween-20) in a 95°C water bath for 20 min, and then solution was allowed to cool to RT. For 6 week old brains 25 m sagittal sections were taken on a Leica CM3050S cryostat and then processed without antigen retrieval. All sections were blocked in blocking buffer (2% normal goat serum, 0.3% Triton X-100, in 1XPBS) for 1 hr at RT and then stained with respective primary antibodies in blocking buffer (1:250 anti-Dnmt3a (VWR 64B1446) and 1:200 anti-Myc (Sigma C3956) for 2 week old brains and 1:500 anti-MeCP2 (Cell signaling 3456) and 1:800 anti-GFP (abcam 13970) for 6 week old brains). Sections were washed 4X in 1XTBST at RT for 5 min each and then stained with secondary antibodies overnight at 4°C (1:1000 goat anti-mouse 555 (Invitrogen A21127) and 1:750 goat anti-rabbit 488 (Bethyl A120-201D2) for 2 week old brains and 1:750 goat anti-chicken 488 (Invitrogen A11039) and 1:1000 donkey anti Rabbit-568 (Invitrogen A10042) for 6 week old brains). Sections were washed 4x in 1XTBST at RT for 5 min each and then stained with 2.5 g DAPI in 1XPBS then washed in 1XPBS, both for 10 min at RT. Sectioned were mounted in VECTASHIELD HardSet Antifade Mounting Medium without DAPI (Vector Laboratories H-1400) and let dry overnight at 4°C. Images were taken on a Zeiss LSM 880 with Airyscan microscope at 20X. Images were taken with the same laser and gain settings and processed equivalently to facilitate comparison across genotypes. Of note, Dnmt3a related brains were processed at 2 weeks of age as Dnmt3a protein levels are at their peak, but then decline soon after this developmental time window (*Lister et al., 2013*) making immunofluorescence challenging at 6 weeks of age.

## Electrophysiology

Acute fresh brain slices were prepared from 6 week old, male mice. Coronal slices (350 m thick) containing striatum were cut with a vibratome (Leica Microsystems Inc, Buffalo Grove, IL) in a chamber filled with cutting solution containing 110 mM $C_5H_{14}ClNO$, 25 mM $NaHCO_3$, 25 mM D-glucose, 11.6 mM $C_6H_7O_6Na$, 7 mM $MgSO_4$, 3.1 mM $C_3H_3NaO_3$, 2.5 KCl, 1.25 mM $NaH_2PO_4$ and 0.5 mM $CaCl_2$. The slices were then incubated in artificial cerebrospinal fluid (ACSF) containing 119 mM NaCl, 26.2 mM $NaHCO_3$, 11 mM D-glucose, 3 mM KCl, 2 mM $CaCl_2$, 1 mM $MgSO_4$, 1.25 mM $NaH_2PO_4$ at RT. The solutions were bubbled with 95% $O_2$ and 5% $CO_2$. Whole-cell recording was made from medium spiny neurons in the dorsal striatum by using a patch-clamp amplifier (MultiClamp 700B, Molecular Devices, Union City, CA) under infrared differential interference contrast optics. Microelectrodes were made from borosilicate glass capillaries and had a resistance of 2.5–4 M. Data was collected with a digitizer (DigiData 1440A, Molecular Devices). The analysis software pClamp10 (Molecular Devices) and Minianalysis 6.0.3 (Synaptosoft Inc, Decatur, GA) were used for data analysis. Miniature IPSCs were recorded in voltage-clamp mode in the presence of 10 M 6-cyano-7-nitroquinoxaline-2, 3-dione (CNQX), 50 M D-2-amino-5-phosphonopentanoic acid (AP5) and 1 M TTX. The glass pipettes were filled with high-Cl-intrapipette solution containing 145 mM KCl, 10 mM HEPES, 2 mM MgCl2, 4 mM MgATP, 0.3 mM Na2GTP and 10 mM Na2-phosphocreatine, pH 7.2 (with KOH). Signals were filtered at 2 KHz and sampled at 10 KHz. Data were discarded when the change in the series resistance was above 20% during the course of the experiment. The whole-cell recording was performed at 25 (±1) °C with the help of an automatic temperature controller (Warner Instruments, Hamden, CT). Data were analyzed with ordinary one-way ANOVA with Tukey's multiple comparisons. Results were considered to be significant at $p < 0.05$.

## Nuclei isolation

Whole striatum was dissected from 6 week old *Dnmt3a*<sup>flox/flox</sup>;*R26-CAG-LSL-Sun1-sfGFP-Myc*<sup>+/-</sup>; *Slc32a1-Cre*<sup>+/-</sup>, *Mecp2*<sup>flox/y</sup>;*R26-CAG-LSL-Sun1-sfGFP-Myc*<sup>+/-</sup>;*Slc32a1-Cre*<sup>+/-</sup>, and *R26-CAG-LSL-Sun1-sfGFP-Myc* <sup>+/-</sup>;*Slc32a1-Cre*<sup>+/-</sup>, male mice in ice-cold HB buffer (0.25M sucrose, 25 mM KCL, 5 mM MgCl2, 20 mM Tricine-NaOH pH 7.8) and flash frozen in liquid nitrogen and stored at 80℃. Experimental striatum (both halves of one mouse) were subjected to nuclear isolation and sorting for a total of 4 animals for each genotype. Individual striatum were dounced one at a time in 9 mL lysis buffer (0.32M Sucrose, 5 mM CaCl$_2$, 3 mM Mg(Ac)$_2$, 0.1 mM EDTA pH8, 10 mM Tris-HCl pH8, 1 mM DTT, 0.1% Triton X-100, 1X protease inhibitors (Sigma 5056489001), ribonuclease inhibitor (Promega N261A) 30 U/ml in DEPC treated water) in a 15 mL dounce homogenizer (VWR 62400–642) 15 strokes with loose pestle, followed by 35 strokes with tight pestle. Homogenized tissue was gently layered onto two ultracentrifuge tubes (4.5 ml on each) filled with 8.5 mL sucrose solution (1.8M sucrose, 3 mM Mg(Ac)$_2$, 1 mM DTT, 10 mM Tris-HCl pH8, 1X protease inhibitors (Sigma 5056489001), ribonuclease inhibitor 30 U/ml in DEPC treated water). Samples were spun in a Beckman Optima LE 80K ultracentrifuge for 2.5 hr at 6℃ with slow deceleration. After centrifugation, the top layer of gradient and mitochondrial layer was discarded carefully with vacuum until ~3 mL remained. Remaining solution was gently poured off and tubes dabbed with a KimWipe being careful to not disturb the nuclear pellet. Nuclei were rehydrated on ice for 45 min in 500 uL of rehydration buffer that was added drop-wise (0.5% BSA, 1X protease inhibitors, ribonuclease inhibitor 30 U/ml in 1X DPBS). Re-hydrated nuclei were pipetted up and down 50X with using a 1 mL filter tip with the end cut to increase clearance size. Matched samples were then re-pooled and dual labeled with mouse anti-NeuN 647 (Millipore MAB377 labeled using Thermo Fisher Scientific A10475) and anti-GFP 488 (Thermo Fisher Scientific A-21311) at 1:300 and 1:1000 ratio, respectively, for 1 hr at 4℃ with gentle mixing. A BD Influx Cell Sorter at the Salk Flow Cytometry Core facility equipped with a 100 micron nozzle tip was used to isolate nuclei. Sheath fluid and pressure was 1X PBS (no Ca2<sup>+</sup>, Mg2<sup>+</sup>) and 18.5 PSI, respectively. Nuclei were first gated based on light scatter properties to exclude debris (forward versus side scatter) then aggregate exclusion gating was applied (forward scatter as well as side scatter pulse width). Finally, nuclei were selected based on anti-NeuN647 and anti-GFP488 labeling. The nuclei fractions were collected at 4℃ using a 1 or two drop purity sort mode and collected into rehydration buffer described above. Nuclei were then spun down at 5,000 rpm for 15 min at 4℃, solution removed until ~50–100 uL of buffer was left, and frozen on dry ice and stored at 80℃ until DNA/RNA extraction for NGS library preparation.

## RNA extraction, NGS library preparation and RNA-seq

RNA was extracted from sorted nuclei using a single cell RNA purification kit from Norgen Biotek according to manufactures instructions and stored at 80C until library preparation at the Genomic and RNA Profiling Core at Baylor College of Medicine. The Genomic and RNA Profiling Core first conducted Sample Quality checks using the NanoDrop spectrophotometer and Agilent Bioanalyzer 2100. Total RNA was quanted by the user using Qubit 2.0 RNA quantitation assay. The NuGEN Ovation RNA-Seq v2 (protocol p/n 7102, kit p/n 7102–08) and the Rubicon ThruPlex DNA-Seq (protocol: QAM-108–002, kit p/n R400428) kits were used for library preparation as follows:

## NuGEN ovation RNA-Seq system v2 protocol

Purified double-stranded cDNA was generated from approximately 5 ng of total RNA and amplified using both oligo d(T) and random primers. Samples were quantified using the NanoDrop ND-2000 spectrophotometer and Qubit 2.0 DNA quantitation assay. One microgram of each sample's ds-cDNA was sheared using the Covaris LE220 focused-ultrasonicator with a 400 bp target size. The sheared samples were quantified using the Qubit 2.0 DNA quantitation assay. The fragment sizes were viewed on the Agilent Bioanalyzer to verify proper shearing.

## Rubicon ThruPlex DNA-Seq library preparation protocol

A double-stranded DNA library was generated from 50 ng of sheared, double-stranded cDNA, preparing the fragments for hybridization onto a flowcell. This is achieved by first creating blunt ended fragments, then ligating stem-loop adapters with blocked 5' ends to the 5' end of the double-stranded cDNA, leaving a nick at the 3' end. Finally, library synthesis extends the 3' end of the

double stranded cDNA and Illumina-compatible indexes are incorporated with five amplification cycles. The fragments are purified using AMPure XP Bead system. The resulting libraries are quantitated using the NanoDrop ND-1000 spectrophotometer and fragment size assessed with the Agilent Bioanalyzer. A qPCR quantitation is performed on the libraries to determine the concentration of adapter-ligated fragments using the Applied Biosystems ViiA7 TM Real-Time PCR System and a KAPA Library Quant Kit.

## Cluster generation by bridge amplification

Using the concentration from the ViiA7 qPCR machine above, 25pM from each equimolarly pooled library was loaded onto five lanes of a high output v4 flowcell (Illumina p/n PE-401–4001) and amplified by bridge amplification using the Illumina cBot machine (cBot protocol: PE_HiSeq_Cluster_-Kit_v4_cBot_recipe_v9.0). PhiX Control v3 adapter-ligated library (Illumina p/n 15017666) is spiked-in at 2% by weight to ensure balanced diversity and to monitor clustering and sequencing performance. A paired-end 100 cycle run was used to sequence the flowcell on a HiSeq 2500 Sequencing System (Illumina p/n FC-401–4003).

## RNA-seq analysis

Adapter sequences were removed from raw sequencing reads using cutadapt (v1.13), trimmed reads were then aligned to reference genome GRCm38 (GENCODE vM15 Primary assembly) using STAR aligner (*Dobin et al., 2013*) (version 2.5.3a) using default parameters. The number of reads aligned within the gene body (from TSS to TES) of each gene was tabulated using FeatureCount (*Liao et al., 2014*) (v1.5.3) (without extension on both ends). Finally, differential gene expression (DEG) analyses on the read counts were performed using DESeq2 (*Love et al., 2014*) (v1.6.2) in R environment. Genes with total read counts less 10 were filtered out from analysis. For analysis of differentially expressed genes split according to gene length as measured from transcription start site to transcription end site. Genes were defined as long or short according to length standards defined in *Gabel et al. (2015)*.

## DNA methylation sequencing

DNA methylome libraries were generated using a modified snmC-seq method adapted for bulk DNA samples and as previous described in *Sabbagh et al. (2018)*. 1% unmethylated lambda DNA (Promega D1521) was added into each sample. Libraries were sequenced using an Illumina HiSeq 4000 instrument. The mapping of DNA methylome reads was done also described in *Sabbagh et al. (2018)*.

Plots of mCH versus mCG in wild-type mice are made with gene body methylation values for percentage mCH or percentage mCG. All genes are plotted as one point per gene. Density contours are plotted with geom_density_2d in R and Pearson correlations were calculated with the cor function. The plots for the methylation differences are calculated with ((mCH in WT) - (mCH in cKO)) or as ((mCG in WT) - (mCG in cKO)).

## Integrative RNA-seq and methylation analysis

Methylation versus gene expression plots were made using running average binning on significantly differentially expressed genes (DEGs, padj <0.01). To obtain bins, genes were first ordered based on their methylation value. Percent methylation is calculated as ((% mCH in cKO) - (% mCH in WT)) or as ((% mCG in cKO) - (% mCG in WT)). Genes were then binned such that each bin contained the same number of genes, with 80 percent overlap between consecutive bins. One point is plotted per bin. For the plots of genes significantly misregulated only in the *Dnmt3a* cKO model, each bin has 25 genes and the window moves by five genes per bin. For plots of genes significantly misregulated only in the *Mecp2* cKO model, as well as the common DEGs, the bin size is 10 genes and the window slides by two each time. The number of bins obtained for DEGs in the *Dnmt3a* cKO model were 126, and 42 for DEGs that were only significant in the *Dnmt3a* cKO model, and common DEGs, respectively. The number of bins obtained for DEGs in the *Mecp2* cKO model were 62, and 42 for DEGs that were only significant in the *Mecp2* cKO model, and common DEGs, respectively. After binning, a univariate linear model was fit to the data with the lm function in R, and the $R^2$ (percentage variance in log2 fold change explained by methylation) was calculated.

To examine the significance of the observed trends in the running average plots, we compared their $R^2$ values to $R^2$ values from 1000 random gene repetitions, with each repetition containing the same number of non-differentially expressed genes (padj >0.01). Gray points represent each iteration of this process, with the original DEG (padj <0.01) values for $R^2$ shown in larger orange, green, or blue points. P value was then computed as $(r+1)/(n+1)$ (*North et al., 2002*); r is number of repetitions where $R^2$ is greater than that in the DEGs, and n is total number of repetitions.

## RNA-seq deconvolution analysis

Deconvolution analysis to assign DEGs specific to medium spiny neurons (MSN) or interneurons of the mouse striatum was done using the PSEA (Population-Specific Expression Analysis) package from R platform (*Kuhn et al., 2011*). Genes that serve as specific markers for each neuron population were selected from single cell RNA-seq data available on the DropViz database (*Saunders et al., 2018*), and used to calculate marker-based reference signal to build gene-wise population specific models. Samples are separated into two groups (*Mecp2* cKO vs. WT and *Dnmt3a* cKO vs. WT) for the population specific differential analysis. We then applied two rounds of standard filtering (*Kuhn et al., 2012*) to select genes with significant population-specific changes. In the first round of filtering, we only kept genes with better model fit ($R^2 > 0.6$) and low noise levels (intercept/mean < 0.5; intercept p > 0.1). In the second round of filtering, we select genes with significant differential expression (p < 0.05).

In order to determine the efficacy of our deconvolution, we compared the results of PSEA and the original DESeq. A hypergeometric test was performed for each of the conditions (*Mecp2* cKO vs. WT and *Dnmt3a* cKO vs. WT – in both MSN and interneurons) with the phyper function in R with lower.tail set to FALSE. For MSNs, out of 849 total DEGs, DESeq called 213 and PSEA called 29 for *Mecp2* cKO mice; 15 of 29 DEGs called by PSEA were found in the original 213 DEGs called by DESeq. Out of 849 total DEGs, DESeq called 722 for *Dnmt3a* cKO mice, and PSEA called 90; 88 of 90 DEGs called by PSEA were found in the original 722 DEGs called by DESeq. The p values for *Mecp2* cKO vs. WT and *Dnmt3a* cKO vs. WT are 0.000409965 and 3.598891e-06, respectively. For interneurons, out of 849 total DEGs, DESeq called 213 for to *Mecp2* cKO mice, and PSEA called 24; 11 of 24 DEGs called by PSEA were found in the original 213 DEGs called by DESeq. Out of 849 total DEGs, DESeq called 722 for to *Dnmt3a* cKO mice, and PSEA called 19; 17 of 19 DEGs called by PSEA were found in the original 722 DEGs called by DESeq. The p values for *Mecp2* cKO vs. WT and *Dnmt3a* cKO vs. WT are 0.00666093 and 0.196558, respectively. As the major population, the significant overlap between DEGs called from DESeq and PSEA from MSN validate the deconvolution. As the minor population of as few as 5% (*Dudman and Gerfen, 2015*; *Kemp and Powell, 1971*), it is unsurprising that the interneuron overlap is borderline significant and insignificant in *Mecp2* cKO vs. WT and *Dnmt3a* cKO vs. WT respectively, as not much contamination is predicted.

The methodology for our re-examined integrative RNA-seq trends is identical to that described in the Materials and methods section above, with removal of 38 interneuron associated DEGs determined from our PSEA analysis (*Supplementary file 3*). The number of bins obtained for *Dnmt3a* cKO were 121, and 39 for DEGs identified in *Dnmt3a* cKO mice only, and DEGs common to both cKO models, respectively. The number of bins obtained for *Mecp2* cKO were 58, and 39 in DEGs identified in *Mecp2* cKO mice only, and DEGs common to both cKO models, respectively.

## Acknowledgements

We are grateful to members of the Zoghbi lab for helpful discussions, H K Yalamanchili and H-H Jeong for assistance with integrative analysis of RNA-seq and methylation data, and VL Brandt for critical comments on the manuscript. We thank Drs. Michael Greenberg and Hume Stroud for assistance with access to and discussion of published single-nuclei RNA-seq data. We thank the Neurovisualization, Neurobehavioral and RNA In Situ Hybridization Cores at the Jan and Dan Duncan Neurological Research Institute at Texas Children's Hospital supported by the BCM-IDDRC (U.S. NIH Grant U54HD083092 to HYZ). We also thank the Research Services Laboratory at the Center for Comparative Medicine at Baylor College of Medicine. This project was supported by NIH/NINDS 5R01NS057819-13 (HYZ), NIH/ NIMH R01MH112763 (MMB and JRE), the Genomic and RNA Profiling Core at Baylor College of Medicine, as well as the Flow Cytometry Core Facility of the Salk Institute funded by NIH-NCI CCSG: P30 014195 (JRE). LAL is a Howard Hughes Medical Institute Fellow

of the Life Sciences Research Foundation. HYZ and JRE are investigators with the Howard Hughes Medical Institute.

## Additional information

### Competing interests
Huda Y Zoghbi: Senior editor, *eLife*. The other authors declare that no competing interests exist.

### Funding

| Funder | Grant reference number | Author |
|---|---|---|
| National Institute of Neurological Disorders and Stroke | 5R01NS057819-13 | Huda Y Zoghbi |
| National Institute of Mental Health | R01MH112763 | Joseph R Ecker<br>M Margarita Behrens |
| Life Sciences Research Foundation | Postdoctoral Research Fellowship | Laura A Lavery |
| National Institutes of Health | U54HD083092 | Huda Y Zoghbi |
| National Cancer Institute | CCSG: P30 014195 | Joseph R Ecker |
| Howard Hughes Medical Institute | | Huda Y Zoghbi<br>Joseph R Ecker |
| Howard Hughes Medical Institute | Fellow of the Life Sciences Research Foundation | Laura A Lavery |

The funders had no role in study design, data collection and interpretation, or the decision to submit the work for publication.

### Author contributions
Laura A Lavery, Huda Y Zoghbi, Conceptualization, Formal analysis, Supervision, Funding acquisition, Validation, Investigation, Visualization, Project administration; Kerstin Ure, Resources, Formal analysis, Investigation; Ying-Wooi Wan, Software, Formal analysis, Supervision; Chongyuan Luo, Joseph R Ecker, Resources, Software, Formal analysis, Supervision, Methodology; Alexander J Trostle, Resources, Software, Formal analysis, Supervision, Methodology, Project administration; Wei Wang, Joseph R Nery, Resources, Supervision, Investigation, Project administration; Haijing Jin, Data curation; Joanna Lopez, Rosa Castanon, Investigation; Jacinta Lucero, Mark A Durham, Formal analysis, Investigation; Zhandong Liu, Software, Formal analysis, Supervision, Methodology; Margaret Goodell, Resources, Software, Formal analysis, Methodology; M Margarita Behrens, Resources, Formal analysis, Supervision, Investigation, Methodology, Project administration

### Ethics
Animal experimentation: Baylor College of Medicine Institutional Animal Care and Use Committee (IACUC, Protocol AN-1013) approved all mouse care and manipulation.

### Decision letter and Author response
Decision letter https://doi.org/10.7554/eLife.52981.sa1
Author response https://doi.org/10.7554/eLife.52981.sa2

## Additional files

### Supplementary files
• Supplementary file 1. Numbers and statistics for all mouse behavioral assays and methylation datasets.

- Supplementary file 2. RNA-seq data and DEGs used in figures.
- Supplementary file 3. Table of interneuron DEGs identified by PSEA analysis.
- Transparent reporting form

### Data availability

DNA methylome data can be accessed through a web browser at http://neomorph.salk.edu/Striatum_Inhibitory_Neuron.php and at the Gene Expression Omnibus database (GEO) at accession number GSE124009. RNA-seq data can be accessed at GEO at accession number GSE123941.

The following datasets were generated:

Luo C Lavery LA Castanon R Nery JR Zoghbi HY Ecker JR2018Loss of non-CpG methylation in inhibitory neurons impairs neural function through a mechanism that partially overlaps with Rett syndromehttps://www.ncbi.nlm.nih.gov/geo/query/acc.cgi?acc=GSE124009NCBI Gene Expression Omnibus, GSE124009     Lavery LA Wan Y Zoghbi HY2018Loss of non-CpG methylation in inhibitory neurons impairs neural function through a mechanism that partially overlaps with Rett syndromehttps://www.ncbi.nlm.nih.gov/geo/query/acc.cgi?acc=GSE123941NCBI Gene Expression Omnibus, GSE123941The following previously published dataset was used:

| Author(s) | Year | Dataset title | Dataset URL | Database and Identifier |
|---|---|---|---|---|
| Stroud H Su SC Hrvatin S Greben AW Renthal W Boxer LD Nagy MA Hochbaum DR Kinde B Gabel HW Greenberg ME | 2017 | Early-life gene expression in neurons modulates lasting epigenetic states | https://www.ncbi.nlm.nih.gov/geo/query/acc.cgi?acc=GSE103214 | NCBI Gene Expression Omnibus, GSE103214 |

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
