## [Decision Letter]

**Acceptance summary:**

Dnmt3a is the sole known writer of mCH in the brain and MeCP2 is the only known reader. Because the timing of mCH accumulation in neurons matches the onset of symptoms in Rett Syndrome, which is caused by loss of function in MeCP2, the relationship between these two chromatin regulators has formed the basis of a current hypothesis of the pathophysiology of this disease. Here the authors set out to test this relationship by knocking out each gene separately and then comparing the molecular and neurological phenotypes. This study is important because it offers a head to head comparison between two proteins linked to DNA methylation, a writer and a reader, and helps the field to understand the link between DNMT3a/DNA methylation/MECP2 and the regulation of gene expression especially in the context of Rett Syndrome.

**Decision letter after peer review:**

Thank you for submitting your article "Losing Dnmt3a dependent methylation in inhibitory neurons impairs neural function by a mechanism impacting Rett syndrome" for consideration by *eLife*. Your article has been reviewed by three peer reviewers, one of whom is a member of our Board of Reviewing Editors, and the evaluation has been overseen by Kate Wassum as the Senior Editor. The following individual involved in review of your submission has agreed to reveal their identity: Zhaolan Zhou (Reviewer #2).

The reviewers have discussed the reviews with one another and the Reviewing Editor has drafted this decision to help you prepare a revised submission.

Summary:

*Dnmt3a* is the sole known writer of mCH in the brain and MeCP2 is the only known reader. Because the timing of mCH accumulation in neurons matches the onset of symptoms in RTT, the relationship between these two chromatin regulators has formed the basis of a current hypothesis of the pathophysiology of RTT. Here the authors set out to test this relationship by knocking out each gene separately with careful controls for genetic background and then comparing the molecular and neurological phenotypes, focusing in on VIATT-positive GABAergic neurons of the CNS. They find more widespread effects on both gene regulation and physiological function with *Dnmt3a* loss compared with MeCP2 loss and conclude that therefore there must be other readers of mCH. For the cases in which the molecular phenotypes do overlap, profiling of mCH and mCG suggests the RTT-relevant commonalities are related to mCH more than mCG, so they suggest this is an argument supporting mCH as important for the phenotypes of RTT that arise in the absence of MeCP2.

Many of the findings are confirmatory of current models for MeCP2 function at mCH residues, but the data also make it clear that there are very likely other neurologically-important mCH reader proteins that are not yet known. These data lay the groundwork for future discoveries and thus will have high impact in the field of neuroscience.

The reviewers have only a few clarifications that they think will substantially strengthen the manuscript.

Essential revisions:

1) The manuscript needs to discuss a few additional possible explanations for the differences between the *Dnmt3a* and MeCP2 phenotypes. First with respect to the neurological phenotypes, is *Slc32a1-Cre* expressed outside the nervous system? This could, for example, contribute to the differences in size between the *Dnmt3a* and MeCP2 cKO mice. Second, this manuscript does not appear to consider the possibility that aberrant phenotypes in *Dnmt3a* cKO mice could arise from non-enzymatic functions of *Dnmt3a*. Is there evidence for the phenotype of a *Dnmt3a* enzymatic mutant? Performing these experiments is beyond the scope of the current study, but if there is evidence for or against this possibility from the literature it should be discussed here.

2) There are presumably at least three classes of *Slc32a1*+ cells in the striatum, specifically the D1+ MSNs, the D2+ MSNs, and a few classes of GABAergic interneurons. The Ecker lab has already shown that interneuron methylation patterns are hidden in bulk lysates from cortex and one would expect the same here since MSNs are so much more numerous than interneurons in striatum. However in the RNAseq data some interneuron-specific genes are likely detected. This disconnect would impact the RNA/DNA methylation comparisons in Figure 4. The authors should discuss and/or bioinformatically address this point.

3) In the third paragraph of the Discussion, the authors describe that DEGs in both up and down directions are independent of gene length. Have the authors examined their relationship to gene body methylation (mCH and/or mCG)? Similarly, do common and specific targets display particular chromatin or genomic features (other than DNA methylation) that may explain why they are regulated differently. Are all methylation changes associated with changes in gene expression? For non-overlapping DEGs, they could be specific to DNMT3a or MeCP2, but could also be differential in-direct effect of *Dnmt3a* and *Mecp2* loss. This possibility should be mentioned in the Discussion. We feel that addition of any further analysis of the gene expression versus methylation data would be a great benefit to readers.

---

## [Author Response]

Essential revisions:1) The manuscript needs to discuss a few additional possible explanations for the differences between the Dnmt3a and MeCP2 phenotypes. First with respect to the neurological phenotypes, is VIATT-Cre expressed outside the nervous system? This could, for example, contribute to the differences in size between the Dnmt3a and MeCP2 cKO mice.

We have added an experiment where by the expression of Beta-galactosidase is dependent in the presence of Cre recombinase in vivo. Using these mice, we preformed X-gal staining on fixed sections of whole E14.5 embryos to determine the expression of our *Slc32a1-Cre* driver throughout the body. We find specific labeling of cells throughout the nervous system, as well as select labeling of cells in the periphery. We have added these data as Figure 1—figure supplement 3, and included them in the Results section of behavioral analysis (subsection “Loss of Dnmt3a or MeCP2 in inhibitory neurons produces overlapping but not identical behavioral phenotypes”) where we have stated the possibility that non-neuronal phenotypes (e.g. low body weight in *Dnmt3a* cKO mice) could be a result of *Dnmt3a* loss in these peripheral cells. A Materials and methods section for X-gal staining has also been added.

Second, this manuscript does not appear to consider the possibility that aberrant phenotypes in Dnmt3a cKO mice could arise from non-enzymatic functions of Dnmt3a. Is there evidence for the phenotype of a Dnmt3a enzymatic mutant? Performing these experiments is beyond the scope of the current study, but if there is evidence for or against this possibility from the literature it should be discussed here.

It is possible that some of the deficits in *Dnmt3a* cKO mice could be due to the loss of *Dnmt3a* protein levels and not enzymatic function. No definitive experiments comparing *Dnmt3a* knockout mice to mice with a knock-in loss of function, catalytically dead mutant of *Dnmt3a* in vivo have been published. We have added a sentence to our Discussion (third paragraph) to include that functions outside of *Dnmt3a* enzymatic activity may impact the observed *Dnmt3a* cKO phenotype, as they cannot be ruled out by our experiments.

2) There are presumably at least three classes of Slc32a1+ cells in the striatum, specifically the D1+ MSNs, the D2+ MSNs, and a few classes of GABAergic interneurons. The Ecker lab has already shown that interneuron methylation patterns are hidden in bulk lysates from cortex and one would expect the same here since MSNs are so much more numerous than interneurons in striatum. However in the RNAseq data some interneuron-specific genes are likely detected. This disconnect would impact the RNA/DNA methylation comparisons in Figure 4. The authors should discuss and/or bioinformatically address this point.

We have attempted to deconvolve our bulk RNA-seq data using Population-Specific Expression Analysis (PSEA) to address this point. Our analysis was able to detect 38 DEGs that could represent gene expression changes in contaminating interneurons. We removed the interneuron candidate DEGs from our analysis in Figure 4 and find that the trends and significance hold as in Figure 4 (see Figure 4—figure supplement 2). These results have been incorporated into our manuscript (subsection “Integrative gene expression and methylation analysis shows mCH and mCG loss contribute to *Dnmt3a* cKO DEGs, and reveals a strong mCH contribution to RTT”).

It is important to note, however, that the efficacy of our deconvolution was determined to be better than random (hypergeometric test – see Materials and methods, subsection “RNA-seq deconvolution analysis”) for medium spiny neurons and close to random for striatal interneurons. This is not surprising given that interneurons only represent at most 5% of the neuron population in the striatum. Given our statistical success with medium spiny neurons (the major population of neurons in the striatum at 95%), we interpret this to mean that our RNA-seq is fairly clean and representative of medium spiny neurons.

3) In the third paragraph of the Discussion, the authors describe that DEGs in both up and down directions are independent of gene length. Have the authors examined their relationship to gene body methylation (mCH and/or mCG)?

We have looked at DEGs in both the up and down direction and their relationship to gene body methylation. We find that DEGs that go up have higher mCH levels than those that go down. We also see that mCG levels are different between up-and down-regulated genes, though the relationship between direction of gene expression change and higher mCG levels was not constant between DEG categories. The statistics have been added to Figure 4 and we highlight this finding in the subsection “Integrative gene expression and methylation analysis shows mCH and mCG loss contribute to *Dnmt3a* cKO DEGs, and reveals a strong mCH contribution to RTT” and Figure 4—figure supplement 1A-B. The remaining pairwise comparisons for Figure 4A-B can be found in Figure 4—figure supplement 1E-F for the reader. We felt adding all of the comparisons to the plots in Figure 4A-B would make the panels unreadable.

Similarly, do common and specific targets display particular chromatin or genomic features (other than DNA methylation) that may explain why they are regulated differently.

This is a very interesting question. Unfortunately, datasets for chromatin features on our specific, isolated neuron population are largely lacking and we do not feel it would be proper to make conclusions from epigenetic marks in whole tissue or other cell populations. Therefore, any possible relationship between chromatin landscape (outside of methylation) and gene expression changes that occur with loss of either or both *Dnmt3a*/MeCP2 needs to be experimentally explored in future studies.

Of note, we have done Gene Set Enrichment Analysis (GSEA), transcription factor motif analysis (multiple tools), and Search Tool for the Retrieval of Interacting Genes/Proteins (STRING) analysis on common or specific DEGs and have not observed any significant pattern that would explain their categorization.

Are all methylation changes associated with changes in gene expression?

From our analysis we do see changes in methylation that are not obviously correlated with changes in gene expression. Specifically, in Figure 4—figure supplement 1C-D we find that the non-DEGs show a decrease in methylation in the *Dnmt3a* cKO compared to wild-type or the *Mecp2* cKO. However, the level of methylation on non-DEGs in wild-type is significantly lower than methylation seen in wild-type animals for DEG genes altered in either model (Figure 4A-B). Therefore, we propose that it is this significantly larger change in methylation level that occurs when *Dnmt3a* is lost that drives gene expression changes we detect. A sentence highlighting this has been added to the subsection “Integrative gene expression and methylation analysis shows mCH and mCG loss contribute to *Dnmt3a* cKO DEGs, and reveals a strong mCH contribution to RTT”.

For non-overlapping DEGs, they could be specific to DNMT3a or MeCP2, but could also be differential in-direct effect of Dnmt3a and Mecp2 loss. This possibility should be mentioned in the Discussion. We feel that addition of any further analysis of the gene expression versus methylation data would be a great benefit to readers.

Thank you, this is definitely a possibility. We have highlighted this possibility in the Discussion.